# CoVAE: Consistency Training of Variational Autoencoders

## Abstract

Variational Autoencoders (VAEs) provide a principled framework for representation learning and generation, but are limited by the use of a single latent distribution, resulting in a trade-off between reconstruction quality and generative performance. This limitation is commonly addressed through multi-stage pipelines that decouple representation learning from generative modeling. We introduce *Consistency Training of Variational AutoEncoders* (CoVAE), a single-stage training framework that builds on the VAE encoder-prior-decoder structure by learning a continuum of latent representations with increasing regularization. The encoder produces noise-controlled latent embeddings governed by a time-dependent regularization parameter, smoothly transitioning from near-deterministic encodings to an isotropic Gaussian prior. The decoder is trained with a consistency-based reconstruction loss that enforces agreement across latent regularization levels while retaining KL-based variational regularization. Rather than optimizing a new likelihood-based ELBO, CoVAE should be understood as a variationally regularized consistency objective over a time-dependent latent path. This formulation enables joint representation learning and generation within a single autoencoding model, without auxiliary priors or multi-stage training. CoVAE generates high-quality samples in one or few decoding steps, significantly outperforming comparable single-stage VAE baselines while maintaining strong reconstruction performance. Overall, CoVAE offers an alternative paradigm for training generative autoencoders through structured, time-dependent latent representations.

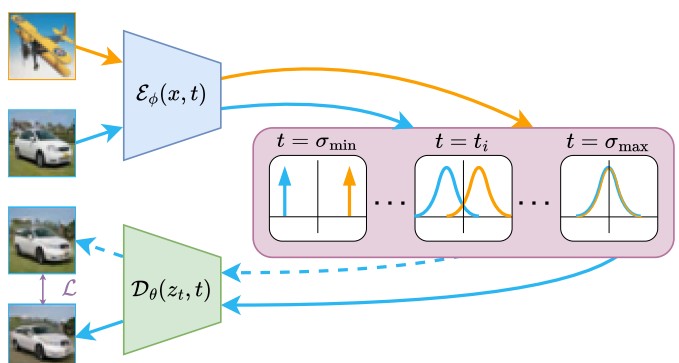 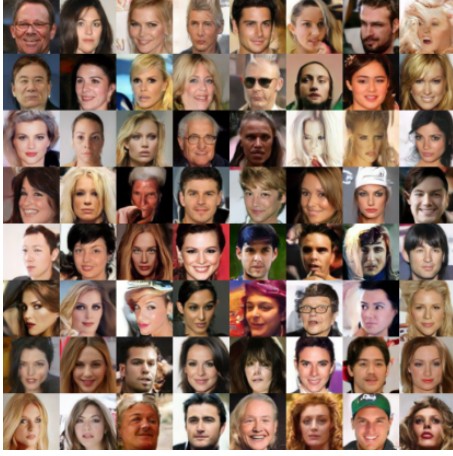

Figure 1: (Left) Schematic representation of CoVAE. The VAE-style model is trained to learn a time-dependent latent distribution, which transitions to Gaussian as time increases. With a loss similar to Consistency Training, the reconstruction at a smaller time steps is used as a target (therefore does not receive gradients, represented with the dashed line) for the prediction at the bigger time steps. (Right) 2-step uncurated samples from CoVAE trained on CelebA 64.

# 1 Introduction

Deep Generative Models (DGMs) are deep neural networks trained to generate samples from an unknown data distribution, generally by learning a mapping between data samples and random noise. Variational Autoencoders (VAEs) (Kingma, 2013; Rezende et al., 2014) provide a principled framework for this goal by jointly learning an encoder and a decoder under a variational objective. Through a shared latent space, VAEs support both representation learning and generation within a single model, making them a natural and widely adopted class of generative autoencoders.

A central limitation of VAEs arises from the use of a *single latent distribution* to support multiple, competing objectives. The reconstruction term favors near-deterministic latent encodings, while the Kullback-Leibler regularization enforces alignment with a simple prior distribution, typically an isotropic Gaussian. The balance between these effects is commonly controlled by a scalar coefficient $\beta$, as in $\beta$-VAEs (Higgins et al., 2017). Small values of $\beta$ yield accurate reconstructions but poorly matched latent priors, while larger values promote stronger prior alignment and more constrained latent representations at the cost of reconstruction fidelity. Despite numerous extensions, including normalizing flows (Rezende & Mohamed, 2015; Kingma et al., 2016; Kobyzev et al., 2020; Papamakarios et al., 2021) and hierarchical latent structures (Sønderby et al., 2016; Maaløe et al., 2019; Child, 2021; Vahdat & Kautz, 2020), this trade-off remains intrinsic to pointwise VAE training objectives.

In practice, this limitation has motivated the widespread use of *two-stage training procedures*. A VAE is first trained with weak regularization to obtain high-quality latent representations, after which a separate and more expressive generative model is trained as a prior over the learned latent space (Razavi et al., 2019; Rombach et al., 2022; Podell et al., 2024; Esser et al., 2024). These approaches have proven highly effective and form the basis of many state-of-the-art generative models. At the same time, the quality and structure of the latent space produced by the autoencoder remain central to the overall performance of such pipelines. A large body of work supports the manifold hypothesis, suggesting that high-dimensional data concentrate near low-dimensional structures (Pope et al., 2021; Brown et al., 2023; Stanczuk et al., 2024; Loaiza-Ganem et al., 2024; Ventura et al., 2025; Bengio et al., 2013). From this perspective, an autoencoder that learns latent representations supporting both faithful reconstruction and meaningful sampling from a simple prior provides a stronger foundation for downstream generative modeling. In particular, strong standalone generative performance can be viewed as an important diagnostic of latent space quality, complementing reconstruction metrics and offering insight into how well the learned representation aligns with the assumed prior, even when a more expressive generative model is subsequently trained in latent space.

In this work, we revisit the training of generative autoencoders from a different perspective. We observe that varying the regularization strength $\beta$ does not merely interpolate between reconstruction- and generation-oriented solutions, but instead induces a *family of valid latent representations* with different levels of information, noise, and prior alignment. Rather than selecting a single operating point, we propose to explicitly learn this continuum of representations within a single model. To this end, we introduce *Consistency Training of Variational AutoEncoders* (CoVAE) (see Figure 1), a single-stage training framework that builds on the VAE encoder–prior–decoder structure by combining time-dependent KL regularization with a consistency-based reconstruction objective. The CoVAE encoder learns a sequence of progressively noised latent representations governed by a monotonically increasing function $\beta(t)$, smoothly transitioning from near-deterministic encodings to latents increasingly aligned with a standard Gaussian prior. Each time step corresponds to a distinct level of latent regularization, making CoVAE similar in spirit to an amortized ensemble of $\beta$-VAEs. Importantly, the time variable indexes latent regimes within the model, rather than a training-time annealing schedule for a single VAE objective. Crucially, instead of applying a pointwise reconstruction loss at each latent level, we replace it with a *consistency reconstruction loss* defined over adjacent regularization levels. Reconstructions obtained at earlier, lightly regularized latent states act as fixed targets for more heavily regularized representations, effectively bootstrapping the decoding process across the latent hierarchy. This training strategy draws inspiration from discrete Consistency Models (Song et al., 2023; Song & Dhariwal, 2024), but differs fundamentally in scope and formulation. CoVAE does not learn a score function, a data-space vector field, or an explicit forward noising process. Instead, the encoder itself defines a learned, time-dependent latent noising mechanism through the reparametrization trick, while the decoder is trained

to be consistent across this latent trajectory. As a result, CoVAE retains the encoder-prior-decoder structure and KL-based variational regularization of VAE-family autoencoders, but should not be interpreted as optimizing a new likelihood-based ELBO. Rather, it defines a variationally regularized consistency objective over a learned time-dependent latent path.

The resulting model is a generative autoencoder that can be trained end-to-end in a single stage, without auxiliary priors or iterative data-space sampling. CoVAE supports one- or few-step generation directly from the latent prior, while maintaining strong reconstruction performance and a structured latent space. Empirically, we show that CoVAE substantially improves generative quality over standard VAE and $\beta$-VAE baselines across multiple image benchmarks, while retaining the advantages of autoencoder-based modeling.

## 2 Background

### 2.1 Variational Autoencoders

In Variational Autoencoders (Kingma, 2013; Rezende et al., 2014), an encoder network $\boldsymbol{\mathcal{E}}_{\boldsymbol{\phi}}(.) : \mathbb{R}^D \to \mathbb{R}^{2d}$ parametrized by $\boldsymbol{\phi}$ is used to learn a mapping from data points $\boldsymbol{x} \sim p_{\text{data}}$ to a probability distribution over a latent space $p(\boldsymbol{z} \mid \boldsymbol{x})$. In the simplest case, this distribution is assumed to be a diagonal Gaussian $q_{\boldsymbol{\phi}}(\boldsymbol{z} \mid \boldsymbol{x}) = \mathcal{N}(\boldsymbol{\mathcal{E}}_{\boldsymbol{\phi}}^{\boldsymbol{\mu}}(\boldsymbol{x}), \boldsymbol{\mathcal{E}}_{\boldsymbol{\phi}}^{\boldsymbol{\sigma}}(\boldsymbol{x})^2 \boldsymbol{I})$, with $\boldsymbol{\mathcal{E}}_{\boldsymbol{\phi}}^{\boldsymbol{\mu}}$ and $\boldsymbol{\mathcal{E}}_{\boldsymbol{\phi}}^{\boldsymbol{\sigma}}$ denoting the mean and standard deviation of the approximate posterior. The encoder is paired with a prior distribution $p(\boldsymbol{z})$ over the latents, commonly a spherical Gaussian, and a decoder network $\boldsymbol{\mathcal{D}}_{\boldsymbol{\theta}}(.) : \mathbb{R}^d \to \mathbb{R}^D$ parametrized by $\boldsymbol{\theta}$ is trained to map latent variables back to the data space. The model is trained by minimizing the variational objective

$$\mathcal{L}_{\text{VAE}}(\boldsymbol{\theta}, \boldsymbol{\phi}, \beta) = \mathbb{E}_{\boldsymbol{x}, \boldsymbol{z}}\left[\|\boldsymbol{\mathcal{D}}_{\boldsymbol{\theta}}(\boldsymbol{z}) - \boldsymbol{x}\|^2 + \beta\, KL\Big(\mathcal{N}(\boldsymbol{\mathcal{E}}_{\boldsymbol{\phi}}^{\boldsymbol{\mu}}(\boldsymbol{x}), \boldsymbol{\mathcal{E}}_{\boldsymbol{\phi}}^{\boldsymbol{\sigma}}(\boldsymbol{x})^2 \boldsymbol{I}) \,\big\|\, \mathcal{N}(\boldsymbol{0}, \boldsymbol{I})\Big)\right], \tag{1}$$

where $\beta$ is a scalar hyperparameter that controls the strength of latent regularization and regulates the trade-off between reconstruction fidelity and prior matching. Latent samples $\boldsymbol{z} \sim q_{\boldsymbol{\phi}}(\boldsymbol{z} \mid \boldsymbol{x})$ are obtained via the *reparametrization trick*, which allows gradients to propagate through the stochastic sampling operation:

$$\boldsymbol{z} = \boldsymbol{\mathcal{E}}_{\boldsymbol{\phi}}^{\boldsymbol{\mu}}(\boldsymbol{x}) + \boldsymbol{\mathcal{E}}_{\boldsymbol{\phi}}^{\boldsymbol{\sigma}}(\boldsymbol{x})\boldsymbol{\epsilon}, \quad \boldsymbol{\epsilon} \sim \mathcal{N}(\boldsymbol{0}, \boldsymbol{I}). \tag{2}$$

In this expression, the mean encoder $\boldsymbol{\mathcal{E}}_{\boldsymbol{\phi}}^{\boldsymbol{\mu}}(\boldsymbol{x})$ defines a deterministic latent embedding of the data point $\boldsymbol{x}$, while $\boldsymbol{\mathcal{E}}_{\boldsymbol{\phi}}^{\boldsymbol{\sigma}}(\boldsymbol{x})$ controls the magnitude of additive Gaussian noise. The effective signal-to-noise ratio $\left\|\boldsymbol{\mathcal{E}}_{\boldsymbol{\phi}}^{\boldsymbol{\mu}}(\boldsymbol{x})\right\|^2 / \left\|\boldsymbol{\mathcal{E}}_{\boldsymbol{\phi}}^{\boldsymbol{\sigma}}(\boldsymbol{x})\right\|^2$ implicitly depends on $\beta$, with small values corresponding to near-deterministic encodings. Increasing $\beta$ encourages the aggregate posterior to better match the prior and is often associated with improved disentanglement in latent space, while smaller values favor accurate reconstruction (Higgins et al., 2017; Burgess et al., 2018). However, selecting a single value of $\beta$ that simultaneously yields high-quality reconstructions and a well-covered latent prior remains challenging. This limitation is intrinsic to the use of a single latent distribution trained with a pointwise reconstruction objective. In particular, mismatches between the aggregate posterior and the prior can give rise to the *prior hole* problem (Hoffman & Johnson, 2016; Rosca et al., 2018), where regions of the prior correspond to latent codes that are not mapped to in-distribution data. A common workaround is to train the autoencoder with weak regularization (small $\beta$) to obtain high-quality representations, and then fit a more expressive generative model as a prior over the learned latent space. While effective, this strategy introduces additional training stages, auxiliary model parameters, and increased sampling cost. Importantly, these complications do not stem from the autoencoder architecture itself, but rather from the challenge of jointly optimizing reconstruction and generative objectives under a single, fixed level of latent regularization.

### 2.2 Diffusion Models

We briefly review diffusion models to highlight structural connections with variational autoencoders and to motivate time-dependent noise injection and denoising-based training objectives. Diffusion models are not

treated here as a modeling target, but rather as a conceptual reference. For a full SDE-based formulation, we refer the reader to Song et al. (2021b).

A diffusion model is defined by a time-dependent corruption process, typically of linear Gaussian form:

$$\boldsymbol{x}_t = \boldsymbol{\mathcal{F}}(\boldsymbol{x}, t) = a_t \boldsymbol{x} + b_t \boldsymbol{\epsilon}, \quad \boldsymbol{\epsilon} \sim \mathcal{N}(\boldsymbol{0}, \boldsymbol{I}), \tag{3}$$

where $a_t$ and $b_t$ are time-dependent scalar functions. As $t$ increases, the corrupted sample $\boldsymbol{x}_t$ becomes progressively noisier, approaching an isotropic Gaussian distribution at a maximum time $T$. Given samples from this forward process, a time-dependent neural network $\hat{\boldsymbol{x}}_{\boldsymbol{\theta}}(.,.) : \mathbb{R}^D \to \mathbb{R}^D$ is trained to predict the original clean sample $\boldsymbol{x}$ from its noisy counterpart $\boldsymbol{x}_t$ by minimizing the denoising score matching objective

$$\mathcal{L}_{\text{DSM}}(\boldsymbol{\theta}) = \mathbb{E}_t \left[ \lambda(t) \mathbb{E}_{\boldsymbol{x}} \left[ \mathbb{E}_{\boldsymbol{x}_t | \boldsymbol{x}} \left[ \| \hat{\boldsymbol{x}}_{\boldsymbol{\theta}}(\boldsymbol{x}_t, t) - \boldsymbol{x} \|^2 \right] \right] \right], \tag{4}$$

where $\lambda(t)$ is a time-dependent weighting function. After training, sampling is typically performed by approximately inverting the forward process through numerical integration of a reverse-time dynamical system (Song et al., 2021a). While this iterative procedure is central to diffusion-based generative modeling, for our purposes the key aspect is the training objective itself: a reconstruction-like loss defined across a continuum of noise levels induced by a known corruption process. From this perspective, diffusion models can be viewed as learning a denoising decoder conditioned on a fixed, non-learned encoder $\boldsymbol{\mathcal{F}}(\boldsymbol{x}, t)$. This interpretation will be useful when relating diffusion-based objectives to autoencoder training and when motivating consistency-based alternatives to pointwise reconstruction losses.

## 2.3 Consistency Models

Consistency Models (CMs) (Song et al., 2023) are a recent alternative to diffusion models designed for one- or few-step generation. In CMs, a time-dependent neural network $\boldsymbol{f}_{\boldsymbol{\theta}}(.,.) : \mathbb{R}^D \to \mathbb{R}^D$ is trained to satisfy a self-consistency constraint across noise levels, without explicitly learning a vector field or requiring numerical integration at sampling time. More specifically, CMs directly learn a mapping from a corrupted input $\boldsymbol{x}_t$ to a clean output $\boldsymbol{x}$, constrained to be invariant along the corruption trajectory. CMs must satisfy two conditions: a boundary condition $\boldsymbol{f}_{\boldsymbol{\theta}}(\boldsymbol{x}, 0) = \boldsymbol{x}$, and a self-consistency condition $\boldsymbol{f}_{\boldsymbol{\theta}}(\boldsymbol{x}_t, t) = \boldsymbol{f}_{\boldsymbol{\theta}}(\boldsymbol{x}_{t'}, t')$, which enforces that corrupted versions of the same data point at different noise levels $t$ and $t'$ map to the same prediction. In practice, these conditions are commonly enforced using the preconditioning proposed by Karras et al. (2022):

$$\boldsymbol{f}_{\boldsymbol{\theta}}(\boldsymbol{x}_t, t) = c_{\text{skip}}(t) \boldsymbol{x}_t + c_{\text{out}}(t) \boldsymbol{F}_{\boldsymbol{\theta}}(\boldsymbol{x}_t, t), \tag{5}$$

where $c_{\text{skip}}(t)$ and $c_{\text{out}}(t)$ are time-dependent scalar functions such that $c_{\text{skip}}(0) = 1$ and $c_{\text{out}}(0) = 0$.

While CMs can be trained using a continuous-time formulation, the continuous objective can be subject to instabilities and requires several technical considerations to work properly (Lu & Song, 2025). A commonly used alternative is a discrete-time training objective:

$$\mathcal{L}_{\text{CM}}^{\text{disc}}(\boldsymbol{\theta}) = \mathbb{E}_{\boldsymbol{x}_t, t} \left[ \lambda(t) \| \boldsymbol{f}_{\boldsymbol{\theta}}(\boldsymbol{x}_t, t) - \boldsymbol{f}_{\boldsymbol{\theta}^-}(\boldsymbol{x}_{t'}, t') \|^2 \right], \tag{6}$$

where $t$ and $t'$ are neighboring noise levels chosen according to the discretization strategy (Song & Dhariwal, 2024; Geng et al., 2025b), and $\boldsymbol{\theta}^-$ denotes a frozen copy of the network parameters. Intuitively, this loss gradually bootstraps the boundary condition at $t = 0$ to higher noise levels by minimizing discrepancies along the corruption trajectory. For our purposes, it is important to note that, due to the boundary condition, the consistency loss reduces to a standard autoencoder reconstruction loss when $t' = 0$:

$$\| \boldsymbol{f}_{\boldsymbol{\theta}}(\boldsymbol{x}_t, t) - \boldsymbol{x} \|^2, \tag{7}$$

where $\boldsymbol{x}_t = a_t \boldsymbol{x} + b_t \boldsymbol{\epsilon}$. This observation suggests that the network $\boldsymbol{f}_{\boldsymbol{\theta}}(\boldsymbol{x}_t, t)$ can be interpreted as a decoder architecture, while the forward corruption process plays the role of a fixed, non-learned encoder.

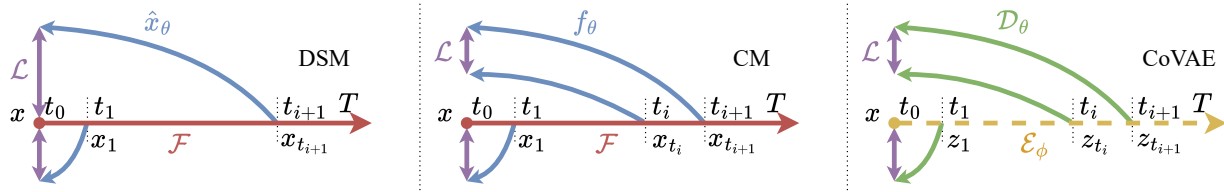

Figure 2: In this figure, we show a diagram of how CoVAE relates to Diffusion and Consistency Models. In diffusion and consistency models, a forward process $\mathcal{F}$ is used to add noise to data as a function of the time parameter $t$. A neural network $\hat{x}_{\theta}$ is trained to predict the clean data in DSM, or $f_{\theta}$ is trained to match predictions across neighboring time steps in discrete CMs using the reconstruction loss $\mathcal{L}$. In CoVAE, the encoder $\mathcal{E}_{\phi}$ is used to obtain a noisy latent state $z_t$ via the reparametrization trick, in analogy to $\mathcal{F}$. The dashed line indicates that latent corruption is induced implicitly through the regularization term $\beta(t)$, rather than by an explicit forward process, and does not correspond to a physical notion of time. The decoder $\mathcal{D}_{\theta}$ then maps latent representations back to data space and is trained using a consistency-based reconstruction loss.

## 3 Method

Before introducing our contribution, we highlight structural similarities between the training objectives discussed in Section 2, as illustrated in Figure 2. In particular, we focus on the role of time-dependent noise injection and reconstruction objectives across noise levels, which provide a useful lens for reinterpreting variational autoencoder training. These connections motivate a reformulation of $\beta$-VAEs with time-dependent regularization and, ultimately, the use of a consistency-based reconstruction loss in latent space.

We first direct the attention to equations 2 and 3, noting that the forward kernel commonly used in diffusion can be seen as a time-dependent version of the reparametrization trick, where drift and diffusion terms are simple, predefined, dimensionality-preserving transformations. On the contrary, VAEs use learned nonlinear dimensionality-reducing mapping as drift and diffusion terms. Note that in DMs the diffusion term does not need to be scalar and can be data dependent (see appendix A from Song et al. (2021b)). Motivated by these observations, one can first consider a time-dependent extension of $\beta$-VAEs:

$$\mathcal{L}_{t-\text{VAE}}(\boldsymbol{\theta}, \boldsymbol{\phi}) = \mathbb{E}_t[\mathcal{L}_{\text{VAE}}(\theta, \phi, \beta(t))], \tag{8}$$

where $\beta(t)$ is a monotonically increasing weighting function. By appropriately defining $\beta(t)$, the learned latents transition between near-deterministic encodings and Gaussian-like latent distributions, and the reparametrization trick can be expressed as

$$z_t = \mathcal{E}_{\boldsymbol{\phi}}^{\mu}(\boldsymbol{x}, t) + \mathcal{E}_{\boldsymbol{\phi}}^{\sigma}(\boldsymbol{x}, t)\boldsymbol{\epsilon}, \quad \boldsymbol{\epsilon} \sim \mathcal{N}(\boldsymbol{0}, \boldsymbol{I}). \tag{9}$$

Equation 9 resembles a learned latent noise-injection process, where the amount of stochasticity and prior alignment is controlled indirectly through the time-dependent KL weight. However, this construction alone is not our contribution. It would still optimize an independent pointwise reconstruction loss at each regularization level, and therefore would simply amortize a family of $\beta$-VAEs within a shared network. Such a model still commits each latent level to its own reconstruction-regularization trade-off, and does not by itself provide a mechanism for transferring reconstruction ability from informative, weakly regularized latents to more prior-aligned latents.

CoVAE differs from this time-dependent VAE baseline in the training objective. Instead of treating each value of $t$ as an independent operating point with its own pointwise reconstruction loss, we couple neighboring latent regimes through a consistency-based reconstruction loss. This turns the time-indexed family of latent distributions into a learned latent path along which the decoder is trained to produce stable predictions.

### 3.1 Consistency-style training of variationally regularized autoencoders

Our main proposal is to replace the pointwise reconstruction term of the time-dependent $\beta$-VAE baseline from Eq. 8 with a latent consistency loss inspired by Eq. 6. Given a time discretization $t_0, t_1, \ldots, t_{N-1} = T$

with $N$ time steps, and imposing the identity function at $t_0$ as boundary condition, we can now define a loss for Consistency Training of Variational AutoEncoders (CoVAEs):

$$\mathcal{L}_{\text{CoVAE}}(\boldsymbol{\theta}, \boldsymbol{\phi}) = \mathbb{E}_{\boldsymbol{x}, \boldsymbol{z}, t_i} \left[ \lambda(t_i) \left\| \boldsymbol{\mathcal{D}}_{\boldsymbol{\theta}}(\boldsymbol{z}_{t_i}, t_i) - \boldsymbol{\mathcal{D}}_{\boldsymbol{\theta}^-}(\boldsymbol{z}_{t_{i-1}}, t_{i-1}) \right\|^2 \right. \tag{10}$$
$$\left. + \beta(t_i) KL(\mathcal{N}(\boldsymbol{\mathcal{E}}_{\boldsymbol{\phi}}^{\mu}(\boldsymbol{x}, t_i), \boldsymbol{\mathcal{E}}_{\boldsymbol{\phi}}^{\sigma}(\boldsymbol{x}, t_i)^2 \boldsymbol{I}) \| \mathcal{N}(\boldsymbol{0}, \boldsymbol{I})) \right],$$

where $\lambda(t)$ is a monotonically decreasing weighting function generally used in CMs, and $\boldsymbol{\theta}^-$ are copies of the model parameters that do not receive gradients. The latent variables $\boldsymbol{z}_{t_i}$ and $\boldsymbol{z}_{t_{i-1}}$ are obtained with the time-dependent reparametrization trick from Eq. 9 using the same random direction $\boldsymbol{\epsilon}$ at both times $t_i$ and $t_{i-1}$.

The CoVAE objective should be interpreted as learning a latent-space consistency map rather than maximizing a likelihood-based ELBO. The encoder induces stochastic latent trajectories indexed by $t$, and the decoder $\boldsymbol{\mathcal{D}}_{\boldsymbol{\theta}}(\boldsymbol{z}_t, t)$ is trained to produce stable predictions along these trajectories. Reconstructions at lightly regularized latent states serve as fixed targets for more strongly regularized states, so decoding information is progressively transferred from informative latents to latents that are closer to the prior. The KL term shapes the latent path by encouraging the high-$t$ representations to align with the Gaussian prior, while the consistency term couples neighboring points along this path. Thus, unlike a time-dependent $\beta$-VAE with independent pointwise reconstruction losses, CoVAE defines a variationally regularized consistency objective over a learned latent path.

Differently from DMs and CMs, where time directly parametrizes an explicit forward noising process, in CoVAE the effect of time is induced implicitly through the weighting functions $\beta(t)$ and $\lambda(t)$. These functions regulate the latent information rate, stochasticity, and degree of prior alignment, rather than a physical diffusion process in data space. We next make this interpretation explicit by relating local consistency along the latent trajectory to a terminal VAE-like reconstruction objective, and by interpreting $\beta(t)$ as a time-dependent rate constraint.

### 3.2  A pathwise interpretation of the CoVAE objective

We provide an interpretation of the CoVAE objective as a variationally regularized path-consistency objective. The goal is not to derive a new tight evidence lower bound for CoVAE. Rather, we show that, under standard boundary and smoothness assumptions, the local consistency loss controls a terminal VAE-like reconstruction objective, while the KL term can be interpreted as imposing time-dependent rate constraints on the learned latent path. This perspective is closely related to the analysis of Variational Consistency Training (VCT) (Silvestri et al., 2025) and to the rate-distortion interpretation of $\beta$-VAEs (Burgess et al., 2018).

**A terminal VAE-like objective.**  Let the CoVAE encoder define a family of approximate posteriors

$$q_{\boldsymbol{\phi}}(\boldsymbol{z}_t \mid \boldsymbol{x}, t) = \mathcal{N}\left( \boldsymbol{\mathcal{E}}_{\boldsymbol{\phi}}^{\mu}(\boldsymbol{x}, t), \boldsymbol{\mathcal{E}}_{\boldsymbol{\phi}}^{\sigma}(\boldsymbol{x}, t)^2 \boldsymbol{I} \right), \tag{11}$$

with samples obtained using a shared noise direction $\boldsymbol{\epsilon}$ across all time steps to ensure trajectory coupling:

$$\boldsymbol{z}_t = \boldsymbol{\mathcal{E}}_{\boldsymbol{\phi}}^{\mu}(\boldsymbol{x}, t) + \boldsymbol{\mathcal{E}}_{\boldsymbol{\phi}}^{\sigma}(\boldsymbol{x}, t)\boldsymbol{\epsilon}, \qquad \boldsymbol{\epsilon} \sim \mathcal{N}(\boldsymbol{0}, \boldsymbol{I}). \tag{12}$$

At the largest time $T$, the intended generative use of the model is to decode latent variables that are closely aligned with the prior. If we define a Gaussian observation model

$$p_{\boldsymbol{\theta}}(\boldsymbol{x} \mid \boldsymbol{z}_T, T) = \mathcal{N}\left( \boldsymbol{x}; \boldsymbol{\mathcal{D}}_{\boldsymbol{\theta}}(\boldsymbol{z}_T, T), \sigma^2 \boldsymbol{I} \right), \tag{13}$$

then the corresponding negative ELBO has the usual VAE-like form, up to constants independent of $\boldsymbol{\theta}$ and $\boldsymbol{\phi}$:

$$\mathcal{J}_T(\boldsymbol{\theta}, \boldsymbol{\phi}) = \frac{1}{2\sigma^2} \mathbb{E}_{\boldsymbol{x}, \boldsymbol{z}_T} \left[ \| \boldsymbol{x} - \boldsymbol{\mathcal{D}}_{\boldsymbol{\theta}}(\boldsymbol{z}_T, T) \|^2 \right] + \mathbb{E}_{\boldsymbol{x}} \left[ \text{KL} \left( q_{\boldsymbol{\phi}}(\boldsymbol{z}_T \mid \boldsymbol{x}, T) \| p(\boldsymbol{z}) \right) \right]. \tag{14}$$

This is not the CoVAE training objective. Instead, it represents the endpoint objective that CoVAE is designed to make useful: accurate decoding from highly regularized, prior-aligned latents.

**Local consistency controls endpoint reconstruction.** Consider a discretization $t_0 < t_1 < \cdots < t_N = T$, and define

$$\boldsymbol{y}_i = \boldsymbol{\mathcal{D}_\theta}(\boldsymbol{z}_{t_i}, t_i). \tag{15}$$

Assume that the small-time endpoint approximately satisfies the boundary condition

$$\boldsymbol{y}_0 \approx \boldsymbol{x}. \tag{16}$$

By adding and subtracting neighboring decoder predictions along the latent path, we form a telescoping sum:

$$\boldsymbol{x} - \boldsymbol{y}_N = (\boldsymbol{x} - \boldsymbol{y}_0) + \sum_{i=1}^{N}(\boldsymbol{y}_{i-1} - \boldsymbol{y}_i). \tag{17}$$

Using Cauchy's inequality, we obtain

$$\|\boldsymbol{x} - \boldsymbol{y}_N\|^2 \le (N+1)\left(\|\boldsymbol{x} - \boldsymbol{y}_0\|^2 + \sum_{i=1}^{N}\|\boldsymbol{y}_i - \boldsymbol{y}_{i-1}\|^2\right). \tag{18}$$

Taking expectations over the data $\boldsymbol{x}$ and the shared noise direction $\boldsymbol{\epsilon}$ yields the pathwise upper bound

$$\mathbb{E}\|\boldsymbol{x} - \boldsymbol{\mathcal{D}_\theta}(\boldsymbol{z}_T, T)\|^2 \le (N+1)\left[\mathbb{E}\|\boldsymbol{x} - \boldsymbol{\mathcal{D}_\theta}(\boldsymbol{z}_{t_0}, t_0)\|^2 + \sum_{i=1}^{N}\mathbb{E}\left\|\boldsymbol{\mathcal{D}_\theta}(\boldsymbol{z}_{t_i}, t_i) - \boldsymbol{\mathcal{D}_\theta}(\boldsymbol{z}_{t_{i-1}}, t_{i-1})\right\|^2\right]. \tag{19}$$

Crucially, the tightness of this bound relies on $\boldsymbol{z}_{t_i}$ and $\boldsymbol{z}_{t_{i-1}}$ being explicitly coupled via the shared noise $\boldsymbol{\epsilon}$; without this coupling, the local consistency terms would compare unrelated latent samples rather than neighboring points along the same stochastic trajectory.

In the continuous-time limit, assuming the encoder and decoder induce a differentiable trajectory $t \mapsto \boldsymbol{\mathcal{D}_\theta}(\boldsymbol{z}_t, t)$, the analogous bound becomes

$$\mathbb{E}\|\boldsymbol{x} - \boldsymbol{\mathcal{D}_\theta}(\boldsymbol{z}_T, T)\|^2 \le 2\,\mathbb{E}\|\boldsymbol{x} - \boldsymbol{\mathcal{D}_\theta}(\boldsymbol{z}_{t_0}, t_0)\|^2 + 2(T - t_0)\int_{t_0}^{T}\mathbb{E}\left\|\frac{d}{dt}\boldsymbol{\mathcal{D}_\theta}(\boldsymbol{z}_t, t)\right\|^2 dt. \tag{20}$$

Thus, minimizing the CoVAE consistency loss can be viewed as minimizing a finite-difference approximation to a pathwise regularizer that keeps the decoded output approximately constant along the learned latent trajectory.

**Time-dependent rate constraints.** The remaining part of the CoVAE objective is the KL regularization term. Following the rate-distortion interpretation of $\beta$-VAEs, the expected KL

$$R(t) = \mathbb{E}_{p_{\text{data}}(\boldsymbol{x})}\left[\text{KL}\left(q_\phi(\boldsymbol{z}_t \mid \boldsymbol{x}, t) \,\|\, p(\boldsymbol{z})\right)\right] \tag{21}$$

can be interpreted as an upper bound on the information transmitted through the latent channel at time $t$. Small values of $R(t)$ correspond to latent distributions close to the prior, while larger values allow more informative latent encodings.

From this perspective, CoVAE can be written as a constrained path-consistency problem:

$$\min_{\boldsymbol{\theta}, \boldsymbol{\phi}} \quad \int_{t_0}^{T} \lambda(t)\mathcal{C}_{\boldsymbol{\theta}, \boldsymbol{\phi}}(t)dt \qquad \text{s.t.} \qquad R(t) \le C(t), \quad \forall t \in [t_0, T], \tag{22}$$

where $\mathcal{C}_{\boldsymbol{\theta}, \boldsymbol{\phi}}(t)$ denotes the local consistency error and $C(t)$ specifies the desired information capacity of the latent representation at time $t$. The Lagrangian relaxation of Eq. 22 is

$$\min_{\boldsymbol{\theta}, \boldsymbol{\phi}} \int_{t_0}^{T} \left(\lambda(t)\mathcal{C}_{\boldsymbol{\theta}, \boldsymbol{\phi}}(t) + \beta(t)R(t)\right) dt, \tag{23}$$

up to constants depending on the prescribed capacity profile $C(t)$. This has the same structure as the CoVAE objective: $\lambda(t)$ weights the local consistency loss, while $\beta(t)$ acts as a time-dependent Lagrange multiplier controlling the information rate and prior alignment of the latent representation.

**Interpretation.** Equations 19–23 suggest the following interpretation. CoVAE does not optimize a standard likelihood-based ELBO. Instead, it learns a time-dependent latent path whose endpoint is regularized toward the prior, and a decoder that is approximately constant along this path. The consistency loss transfers decoding information from lightly regularized, informative latents to strongly regularized, prior-aligned latents. The KL term controls the rate of the latent channel across time. Consequently, $\beta(t)$ should not be understood merely as a training-time annealing schedule for a single VAE objective, but as a continuous parameter defining the varying latent rate constraints along which consistency is enforced.

### 3.3 Boundary conditions in latent space

While imposing the identity function at $t_0$ is enough to respect the initial condition required by CMs, we found in practice that using such a simple parametrization can lead to instabilities during training (see Appendix A.7). Inspired by CMs, we aim to incorporate an EDM-style parametrization from equation 5, which cannot be directly used in our settings, as the latent variable $z_t$ and the output of the decoder generally live in spaces with different dimensionality. We propose a different parametrization where instead of using the noisy state $x_t$ (or latent state $z_t$ in CoVAE), we use a learned approximation of the average decoder $\mathbb{E}[x \mid z_t]$. The average decoder provides a faithful reconstruction for small $t$, while yielding increasingly blurred reconstructions as $t$ increases. This quantity is unknown but can be obtained by training a neural network $\hat{x}_\theta$ with the time-dependent VAE loss from equation 8. Note that we use on purpose the same notation $\hat{x}_\theta$ as for DSM, to stress that the role of the network is the same, i.e. to recover the clean data from the latent $z_t$ or noisy state $x_t$. The decoder parametrization becomes as follows:

$$\mathcal{D}_\theta(z_t, t) = c_{\text{skip}}(t)\hat{x}_{\theta^-}(z_t, t) + c_{\text{out}}(t)r_\theta(z_t, t), \tag{24}$$

where $r_\theta$ models the residual of the average decoder network. Note how the parameters of the average decoder network are frozen and do not receive gradients. Instead, the reconstruction part of the CoVAE loss is modified with an additional denoiser-style loss:

$$\mathcal{L}_{\text{CoVAE}}^{\text{rec}} = \mathbb{E}_{x,z,t_i}\left[\lambda(t_i)\left(\left\|\mathcal{D}_\theta(z_{t_i}, t_i) - \mathcal{D}_{\theta^-}(z_{t_{i-1}}, t_{i-1})\right\|^2 + \lambda_{\text{d}}(t_i)\left\|\hat{x}_\theta(z_{t_i}, t_i) - x\right\|^2\right)\right], \tag{25}$$

where $\lambda_{\text{d}}(.)$ is another time dependent weighting function used to regulate the interplay between consistency and denoising loss. In practice, we double the output channels of the decoder, and use half for $\hat{x}_\theta$ and the other half for $r_\theta$, resulting in a negligible increase in model parameters and compute. This is motivated by the fact that the weights of the denoiser network are generally used as initialization for training CMs (Geng et al., 2025b; Lu & Song, 2025), which suggests a certain degree of compatibility between the features needed for the two losses. The identity function at $t_0$ is still applied.

### 3.4 Training and sampling with CoVAE

We report the pseudocode for the CoVAE loss in Algorithm 1.

After training, the model can be used to generate data by decoding samples from the prior, as in standard VAEs. In addition, CoVAEs can optionally leverage the learned time-dependent latent mapping to perform multi-step sampling by re-encoding intermediate samples, adding noise via the reparametrization trick, and re-denoising (see Algorithm 2). For a two-step sampling procedure, three function evaluations are required (two decoder evaluations and one encoder evaluation). We report the design choices used for training CoVAE in Appendix A, and describe alternative formulations in Appendix C.

## 4 Experiments

In this section, we report experimental results on common image benchmarks. We use Fréchet Inception Distance (FID) (Heusel et al., 2017) as an evaluation metric, both on $50k$ samples generated by the models and on encoded–decoded training images using the full dataset, to assess generative and reconstruction performance. For CoVAE, we use $t = \sigma_{\min}$ to compute the reconstruction FID. We provide additional visualizations and samples in Appendix B.

---

**Algorithm 1** CoVAE Loss

---

**Input:** data distribution $p_{\text{data}}$; decoder parameters $\boldsymbol{\theta}$; encoder parameters $\boldsymbol{\phi}$; weighting functions $\lambda(\cdot)$, $\beta(\cdot)$, $\lambda_{\text{d}}(\cdot)$; time-step distribution $p(t)$
Sample $\boldsymbol{x} \sim p_{\text{data}}$
Sample $t_i \sim p(t)$
Sample $\boldsymbol{\epsilon} \sim \mathcal{N}(\boldsymbol{0}, \boldsymbol{I})$
$\boldsymbol{z}_{t_i} \leftarrow \boldsymbol{\mathcal{E}}_{\boldsymbol{\phi}}^{\boldsymbol{\mu}}(\boldsymbol{x}, t_i) + \boldsymbol{\mathcal{E}}_{\boldsymbol{\phi}}^{\boldsymbol{\sigma}}(\boldsymbol{x}, t_i)\boldsymbol{\epsilon}$
$\boldsymbol{z}_{t_{i-1}} \leftarrow \boldsymbol{\mathcal{E}}_{\boldsymbol{\phi}^-}^{\boldsymbol{\mu}}(\boldsymbol{x}, t_{i-1}) + \boldsymbol{\mathcal{E}}_{\boldsymbol{\phi}^-}^{\boldsymbol{\sigma}}(\boldsymbol{x}, t_{i-1})\boldsymbol{\epsilon}$
$\mathcal{L}_{\text{CoVAE}}^{\text{d}} \leftarrow \|\hat{\boldsymbol{x}}_{\boldsymbol{\theta}}(\boldsymbol{z}_{t_i}, t_i) - \boldsymbol{x}\|^2$
$\mathcal{L}_{\text{CoVAE}}^{\text{cm}} \leftarrow \left\|\boldsymbol{\mathcal{D}}_{\boldsymbol{\theta}}(\boldsymbol{z}_{t_i}, t_i) - \boldsymbol{\mathcal{D}}_{\boldsymbol{\theta}^-}(\boldsymbol{z}_{t_{i-1}}, t_{i-1})\right\|^2$
$\mathcal{L}_{\text{CoVAE}}^{\text{kl}} \leftarrow \text{KL}\Big(\mathcal{N}(\boldsymbol{\mathcal{E}}_{\boldsymbol{\phi}}^{\boldsymbol{\mu}}(\boldsymbol{x}, t_i), \boldsymbol{\mathcal{E}}_{\boldsymbol{\phi}}^{\boldsymbol{\sigma}}(\boldsymbol{x}, t_i)^2\boldsymbol{I}) \,\big\|\, \mathcal{N}(\boldsymbol{0}, \boldsymbol{I})\Big)$
$\mathcal{L}_{\text{CoVAE}} \leftarrow \lambda(t_i)\big[\mathcal{L}_{\text{CoVAE}}^{\text{cm}} + \lambda_{\text{d}}(t_i)\mathcal{L}_{\text{CoVAE}}^{\text{d}}\big] + \beta(t_i)\mathcal{L}_{\text{CoVAE}}^{\text{kl}}$
**Output:** $\mathcal{L}_{\text{CoVAE}}$

---

**Algorithm 2** Multistep CoVAE Sampling

---

**Input:** decoder $\boldsymbol{\mathcal{D}}_{\boldsymbol{\theta}}$; encoder $\boldsymbol{\mathcal{E}}_{\boldsymbol{\phi}}$; sequence of time points $\tau_1 > \tau_2 > \cdots > \tau_{N-1}$
Sample $\boldsymbol{\epsilon} \sim \mathcal{N}(\boldsymbol{0}, \boldsymbol{I})$
$\boldsymbol{x} \leftarrow \boldsymbol{\mathcal{D}}_{\boldsymbol{\theta}}(\boldsymbol{\epsilon}, \tau_1)$
**for** $n = 2$ **to** $N-1$ **do**
    Sample $\boldsymbol{\epsilon} \sim \mathcal{N}(\boldsymbol{0}, \boldsymbol{I})$
    $\boldsymbol{z}_{\tau_n} \leftarrow \boldsymbol{\mathcal{E}}_{\boldsymbol{\phi}}^{\boldsymbol{\mu}}(\boldsymbol{x}, \tau_n) + \boldsymbol{\mathcal{E}}_{\boldsymbol{\phi}}^{\boldsymbol{\sigma}}(\boldsymbol{x}, \tau_n)\boldsymbol{\epsilon}$
    $\boldsymbol{x} \leftarrow \boldsymbol{\mathcal{D}}_{\boldsymbol{\theta}}(\boldsymbol{z}_{\tau_n}, \tau_n)$
**end for**
**Output:** $\boldsymbol{x}$

---

## 4.1 MNIST

As a simple benchmark, we compare an equivalent VAE, $\beta$-VAE, and CoVAE on MNIST (Deng, 2012), where we train models with a $7 \times 7$ latent space and one channel ($16\times$ compression rate). The models are trained for $400k$ iterations with batch size 128 and EMA rate 0.9999. CoVAE is trained with the hyperparameters described in Appendix A. From the results in Table 1, we observe that CoVAE achieves significantly improved performance without requiring a trade-off between generation and reconstruction, as in $\beta$-VAEs, confirming the benefits of the bootstrapped, consistency-based time-dependent objective. To further analyze the behavior of the learned time-dependent latent representation, Figure 3 shows a 2D t-SNE visualization (Van der Maaten & Hinton, 2008) of latent embeddings from $10k$ images sampled from the training set at different time steps, using a consistent per-sample noise mask across time. At small time steps, samples from each class are embedded in well-separated regions, while they gradually become more dispersed as time increases. Additional latent-space visualizations are reported in Appendix B.1, Figure 7. Similarly, Figure 4 reports the Signal-to-Noise Ratio (SNR) of the learned latent space averaged over the same $10k$ image embeddings. The average SNR transitions from a large value corresponding to near-deterministic encoding to values close to zero, indicating the approach to pure noise in latent space.

## 4.2 CIFAR-10

We use CIFAR-10 (Krizhevsky et al., 2009) to assess the generative performance of CoVAE, as it is a common image benchmark for DGMs. We refer the reader to Appendix A for a description of all hyperparameters used, and to Appendix A.7 for an ablation study over different CoVAE configurations, including a comparison with and without the boundary conditions from Equation 24. In the following experiments, we train our models using a 112M-parameter configuration, meaning that the decoder has roughly the same number of parameters as architectures commonly used in CMs for CIFAR-10, and a batch size of 1024. For both the baselines and CoVAE, we use a 1024-dimensional latent space. To further improve generative performance,

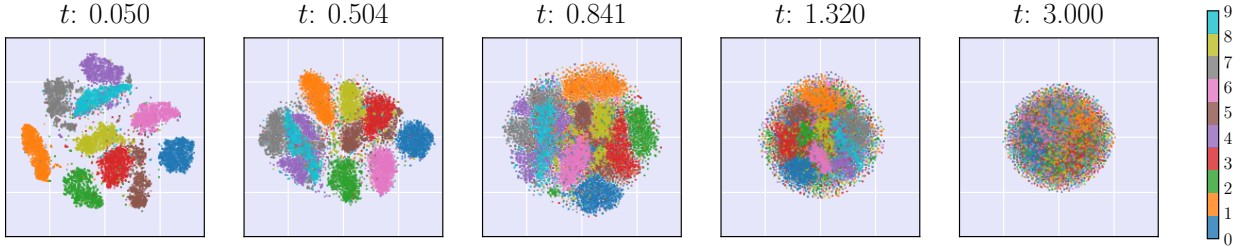

Figure 3: t-SNE embedding of samples from the encoded latents for $10k$ MNIST images at different time steps, with consistent per-sample noise mask across the time steps $t$.

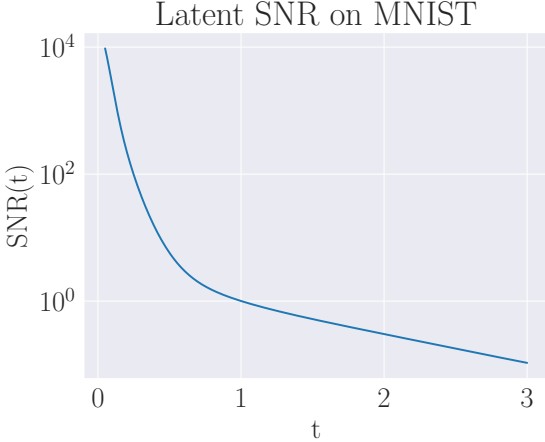

Figure 4: SNR of the latent space for the trained CoVAE model over the different time steps.

| FID ($\downarrow$) on MNIST | | | |
|---|---|---|---|
| | 1 step | 2 steps | Rec. |
| VAE | 17.2 | - | 21.17 |
| $\beta$-VAE ($\beta = 0.5$) | 13.24 | - | 16.56 |
| CoVAE (ours) | **5.62** | **3.83** | **2.19** |

Table 1: FID results (lower is better) for CoVAE and VAE baselines on MNIST.

we optionally train CoVAE with a patch-based adversarial loss $\mathcal{L}_{adv}$, following Esser et al. (2021); Rombach et al. (2022). In Table 2, we report results obtained with our model, compared to VAE and $\beta$-VAE baselines, as well as NVAE (Vahdat & Kautz, 2020) and DC-VAE (Parmar et al., 2021). These baselines are selected as representative single-stage, VAE-based methods that do not rely on training a separate prior for sampling. CoVAE significantly outperforms the equivalent VAE and $\beta$-VAE baselines, and also outperforms both NVAE and DC-VAE; the additional adversarial loss further improves both generative and reconstruction performance. Two-step samples from CoVAE with $\mathcal{L}_{adv}$ are shown in Figure 5.

| FID ($\downarrow$) on CIFAR-10 | | | |
|---|---|---|---|
| Model | 1 step | 2 steps | Rec. |
| VAE | 96.09 | – | 60.76 |
| $\beta$-VAE ($\beta = 0.1$) | 66.79 | – | 30.23 |
| NVAE | 23.49 | – | 2.67 |
| DC-VAE | 17.9 | – | 21.4 |
| CoVAE (ours) | 17.21 | 14.06 | 2.36 |
| CoVAE w/ $\mathcal{L}_{adv}$ (ours) | **11.69** | **9.82** | **2.15** |

| FID ($\downarrow$) on CelebA 64 | | | |
|---|---|---|---|
| Model | 1 step | 2 steps | Rec. |
| NVAE | 14.74 | – | – |
| DC-VAE | 19.9* | – | 14.3* |
| CoVAE (ours) | 10.4 | 9.4 | 5.67 |
| CoVAE w/ $\mathcal{L}_{adv}$ (ours) | **8.27** | **7.15** | **4.90** |

*DC-VAE reports results on $128 \times 128$ resolution.

Table 2: FID ($\downarrow$) on CIFAR-10 (left) and CelebA 64 (right). Lower is better. "Rec." is Reconstruction FID.

### 4.3 CelebA 64 and image manipulation

We further evaluate CoVAE on CelebA (Liu et al., 2015) resized to $64 \times 64$, which is a common benchmark for VAE-based methods. Here, we use CoVAE with a $\times 3$ compression rate, corresponding to a latent space of 4096 dimensions. We report the FID results in Table 2. CoVAE achieves high sample quality and reconstruction performance compared to the baselines, with representative samples shown in Figure 1. Similarly to other VAE-based models, the latent space learned by CoVAE can be used for image manipulation. However, in our case, the availability of a time-dependent latent space allows one to trade off between faithful reconstruction and latent-space disentanglement. We illustrate the effect of latent-space interpolation at different time steps in Figure 5, and provide a more comprehensive analysis in Appendix B.

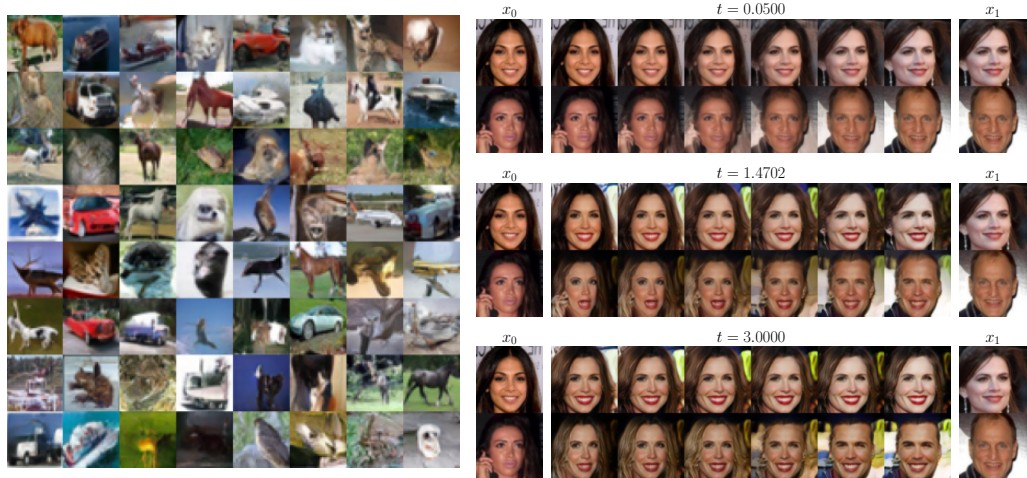

Figure 5: (Left) 2-step samples from CIFAR-10 with CoVAE w/ $\mathcal{L}_{\mathrm{adv}}$. (Right) Latent interpolation on CelebA with different interpolation strengths.

## 5 Related Work

**Single- and few-step generative models:** Recently, many single- or few-step generative models that retain much of the generative performance of diffusion models have emerged, including Consistency Models (Song et al., 2023; Song & Dhariwal, 2024; Geng et al., 2025b; Lu & Song, 2025; Dao et al., 2025), Shortcut Models (Frans et al., 2025), Inductive Model Matching (Zhou et al., 2025), and MeanFlow (Geng et al., 2025a). These approaches have also been used as priors for pretrained VAEs, further reducing sampling time while retaining comparable generative performance. With our method, we aim to show that, despite the success of two-stage training procedures, it is possible to obtain a competitive generative autoencoder with VAE-style encoder–prior–decoder structure using a single training stage with simple prior and posterior distributions. From this perspective, CoVAE can be viewed as integrating consistency-based training into a variationally regularized latent-variable autoencoder framework, without relying on a separately trained prior.

**Learning the forward process:** Several works have explored learning data-dependent forward processes in ambient space. For instance, Nielsen et al. (2024) use an encoder to parameterize the noise injection, Bartosh et al. (2024a) learn the drift and diffusion terms of the SDE directly, and Bartosh et al. (2024b) further extend this approach by combining invertible flows with diffusion to enable exact likelihoods. Unlike these methods, which still rely on separate score models or iterative sampling in data space, CoVAE learns a progressive noising process directly in latent space and jointly trains encoding, noising, and decoding within a single variationally regularized autoencoder using a consistency-based objective. Other methods (Pooladian et al., 2023; Liu et al., 2023; Lee et al., 2023; Albergo et al., 2024; Li et al., 2024; Silvestri et al., 2025) implicitly modify the forward process by introducing couplings between data and noise. Among these,

Lee et al. (2023); Silvestri et al. (2025) use neural networks to learn data–noise couplings in a VAE-style formulation, showing similarities to CoVAE, particularly Silvestri et al. (2025), which applies this idea in the context of consistency models. However, these approaches remain restricted to the ambient space, whereas CoVAE jointly learns both the latent representation and a latent-space forward process, including the latent-to-noise coupling.

**Using a time-dependent VAE:** Some recent works employ time-dependent VAE architectures related to ours. In particular, Batzolis et al. (2023) use a time-dependent encoder in combination with a pretrained score model that serves as a decoder, yielding an improved VAE formulation but still requiring iterative sampling as in diffusion models. Similarly, Uppal et al. (2025) use a time-dependent $\beta$-VAE regularized to obtain a latent space that transitions toward an isotropic Gaussian as time increases. However, they subsequently train a nonlinear diffusion model in the learned latent space, resulting in a two-stage training procedure and requiring multiple sampling steps. In contrast, CoVAE uses the time-dependent latent path directly for generation by training the decoder with a neighboring-time consistency objective, enabling one- or few-step sampling without an auxiliary latent prior. CoVAE is also distinct from training-time $\beta$ schedules and capacity-control methods, such as KL annealing or the capacity-based $\beta$-VAE objective (Burgess et al., 2018; Fu et al., 2019). In those methods, the schedule changes which single operating point is learned during optimization. In CoVAE, $t$ indexes latent regimes evaluated within the trained model, and neighboring regimes are explicitly coupled through a shared consistency-trained decoder. Thus, $\beta(t)$ is not used merely as an annealing schedule, but as a way to define a continuum of latent regularization levels along which consistency is enforced.

# 6 Discussion

In this work, we introduced CoVAE, a single-stage training framework for generative autoencoders that combines KL-based variational regularization with a consistency-based decoder loss. This formulation enables high-quality one- or few-step sampling from a simple prior while maintaining strong reconstruction performance. We presented a set of practical design choices for training CoVAE and demonstrated improved generative and reconstruction results over standard VAE baselines on common image benchmarks.

**Limitations and future work:** While CoVAE shows promising empirical results, several limitations and directions for future work remain. CoVAE is not intended as a likelihood estimator and does not optimize a standard evidence lower bound. Instead, it defines a variationally regularized consistency objective over a learned latent path, trading the clean likelihood semantics of VAEs for an objective aimed at improving one- or few-step generation from prior-aligned latents. Consequently, direct likelihood evaluation remains non-trivial. In addition, the proposed training strategy relies on empirically chosen hyperparameters, such as weighting functions and discretization schemes; deriving more principled formulations could simplify training and further improve performance. A promising direction is to further formalize CoVAE through a pathwise interpretation of the consistency loss, relating local consistency along the latent trajectory to endpoint reconstruction under prior-aligned latents. The architectures used in this work are similar to those commonly adopted in latent diffusion models (Rombach et al., 2022). Recent studies (Skorokhodov et al., 2025; Chen et al., 2025) indicate that architectural refinements can significantly enhance generative quality, efficiency, and compressibility, suggesting that architectures specifically tailored to CoVAE may yield further gains. Moreover, recent advances in training consistency models, including improved initialization (Geng et al., 2025b), truncated training schedules (Lee et al., 2025), and continuous-time formulations (Lu & Song, 2025), may be directly applicable within the CoVAE framework. Finally, an important direction for future work is the integration of CoVAE into latent generative pipelines. Given its improved standalone generative performance and structured latent space, CoVAE may serve as a stronger autoencoder component for two-stage approaches, such as latent diffusion or consistency-based priors trained in latent space. More broadly, the close relationship between CoVAEs and VAEs suggests that established techniques, such as structured priors and hierarchical latent representations (Vahdat & Kautz, 2020), can be naturally extended to CoVAE by introducing time conditioning.

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

## A    Experimental Details

### A.1    Design choices for training CoVAE

Training CoVAE requires a number of design choices which we discuss here. Similarly to discrete CMs, we have to choose discretization strategy, timestep distribution, weighting functions $\lambda(t)$, $\beta(t)$ and $\lambda_{\mathrm{d}}(t)$, the scalar functions $c_{\mathrm{in}}(t)$, $c_{\mathrm{skip}}(t)$ and $c_{\mathrm{out}}(t)$, and minimum and maximum time values $\sigma_{\min}$ and $\sigma_{\max}$. For the discretization strategy, we reuse the discretization introduced in Karras et al. (2022) and used in Song et al. (2023) for consistency training:

$$t_i = \left( \sigma_{\min}^{1/\rho} + \frac{i-1}{N(k)-1} \left( \sigma_{\max}^{1/\rho} - \sigma_{\min}^{1/\rho} \right) \right)^{\rho}, \tag{26}$$

where $\rho$ is a scalar hyperparameter controlling the "linearity" of the discretization ($\rho = 1$ results in a linear discretization, while increasing $\rho$ transitions towards logarithmic), $k \in [0, K]$ is the current training iteration, $K$ is the total training iterations, $i \in [1, N(k)]$ is the discretization step and $N(k)$ is a discretization

curriculum returning the number of discretization steps at the current iteration. As $N(k)$ we choose to use the exponential curriculum from Song & Dhariwal (2024):

$$N(k) = \min\left(s_0 2^{\lfloor \frac{k}{K'} \rfloor}, s_1\right) + 1, \quad K' = \left\lfloor \frac{K}{\log_2 \lfloor s_1/s_0 \rfloor + 1} \right\rfloor, \tag{27}$$

where $s_0 = 2$ and $s_1 = 256$ are initial and final number of steps respectively. During training, we sample time steps uniformly from the given discretization. In CMs, $t_1 = \sigma_{\min}$ is the minimum value that the time steps can assume, and the boundary conditions impose the identity at $t_1$. In CoVAE, we additionally add $t_0 = 0$ to the time steps and apply the boundary condition at $t_0$. This allows us to choose exactly $\sigma_{\min}$ as the first time step used by the encoder (while in CMs $\sigma_{\min}$ is never actually used as the boundary condition applies). After an initial tuning phase (details in A.4), we choose $\sigma_{\min} = 0.05$, $\sigma_{\max} = 3$, $\rho = 7$, $\beta(t) = t^2$ and $\lambda(t) = 1/t$. We always set $c_{\mathrm{skip}}(t) = 1$ (using the derivation in section C.1.3 as a guideline), $c_{\mathrm{in}}(t) = 1$, while for $c_{\mathrm{out}}(t)$ and $\lambda_{\mathrm{d}}(t)$, we use linear interpolations:

$$c_{\mathrm{out}}(t) = \frac{t - \sigma_{\min}}{\sigma_{\max} - \sigma_{\min}}, \quad \lambda_{\mathrm{d}}(t) = c_{\mathrm{d}} + (1 - c_{\mathrm{d}})\left(1 - \frac{t - \sigma_{\min}}{\sigma_{\max} - \sigma_{\min}}\right), \tag{28}$$

where $c_{\mathrm{d}} = 0.1$ is used to reduce the effect of the average decoder loss as $t$ increases, to avoid conflict with the consistency loss. These linear weights are chosen empirically as we do not have an analytic expression for the SNR. The time $t$ is processed as $\log(t)/4$ like in Karras et al. (2022). For the decoder loss, we use the pseudo-huber loss defined like in Song & Dhariwal (2024), while for the average decoder we use the L2 loss. We train all the models for $400k$ iterations with Exponential Moving Average of the weights (EMA), with rate $\mu_{\mathrm{EMA}} = 0.9999$. During training, we make sure that the network uses the same dropout mask for target and prediction computation, as commonly done in CMs. As architecture, we reuse DDPM++ (Ho et al., 2020; Karras et al., 2022) which is based on U-Net (Ronneberger et al., 2015), but without the skip connections between different latent resolutions, effectively obtaining a non-hierarchical time-dependent VAE architecture, where the latent dimensionality is defined by the spatial resolution and channels after encoding. We reuse also the middle block and $1 \times 1$ convolution from Rombach et al. (2022). Following Lu & Song (2025), we replace the Adaptive Group Normalization with Adaptive Double Normalization. The latent size is defined by an the number of channels $z_{\mathrm{ch}}$ multiplied by the spatial resolution after the encoder. The encoder reduces spatial dimensionality by 8 for MNIST and CIFAR-10, and by 16 for CelebA 64.

## A.2 Adversarial Loss

The adversarial loss is used only after $k_w$ warm-up steps, which we set to be half of the training iterations, and is multiplied by a scaling factor:

$$\lambda_{\mathrm{adv}}(k, t) = \lambda(t) \frac{k - k_w}{K - k_w} * \lambda_{\mathrm{adv}} * \mathbb{I}\left(t > \frac{\sigma_{\max}(k - k_w)}{K - k_w}\right), \tag{29}$$

where $k$ is the current training iteration, $\lambda(t)$ is the same time weighting function used for the CoVAE loss, $\lambda_{\mathrm{adv}} = 0.05$ is a constant hyperparameter, and $\mathbb{I}(.)$ is a gate function which applies the adversarial loss to time steps progressively as the iterations increase. The rationale behind the gating is that smaller time steps are better approximations of the data earlier during training.

## A.3 Network configurations

In section A.7 we use different configurations for the neural network in the ablation for different network size. The network differ for the channel multipliers and number of residual blocks as follows:

- Model with 35.8M parameters: Channel multiplyers $= [2, 2, 2]$, resdual blocks $= 2$;

- Model with 54.2M parameters: Channel multiplyers $= [2, 2, 2]$, resdual blocks $= 4$;

- Model with 94M parameters: Channel multiplyers $= [2, 2, 4]$, resdual blocks $= 2$.

We report in table 3 the hyperparameters used for the models in section 4. All our models are trained with precision BFloat16. For training, we use the random seed 42, while for evaluation we set it to 32.

| Model Setups | MNIST | CIFAR-10 | CelebA 64 |
|---|---|---|---|
| Model Channels | 64 | 128 | 128 |
| N° of ResBlocks | 2 | 4 | 2 |
| Attention Resolution | [14] | [16,8] | [16,8] |
| Channel multiplyer | [2, 2, 2] | [2, 2, 4] | [1, 2, 2, 4] |
| Model capacity | 8M | 112M | 81.7M |
| Latent size | 49 | 1024 | 4096 |
| Discriminator Capacity | - | 3M | 3.7M |
| Encoder GFLOPs | 1.4 | 17.4 | 15.0 |
| Decoder GFLOPs | 2.5 | 25.3 | 37.1 |
| **Training Details** | | | |
| Minibatch size | 128 | 1024 | 800 |
| Batch per device | 128 | 512 | 400 |
| Iterations | 400k | 400k | 400k |
| Dropout probability | 20% | 20% | 20% |
| Optimizer | RAdam | RAdam | RAdam |
| Learning rate | 0.0001 | 0.0001 | 0.0001 |
| EMA rate | 0.9999 | 0.9999 | 0.9999 |
| Gradient clip value | 200 | 200 | 200 |
| Number of GPUs | 1 | 2 | 2 |
| GPU types | A100 | H100 | H100 |

Table 3: Model configurations and training details for CoVAE for the different datasets.

## A.4 Initial tuning phase

To find a suitable set of hyperparameters for CoVAE, we use the architecture with 35.8M parameters with batch size 128 and $3\times$ compression rate on CIFAR-10. In early experiments we used $\lambda(t) = 1/t$ and $\beta(t) = t$, but we found the reconstruction loss to become unstable for small value of t. We therefore changed $\beta(t) = t^2$ to allow for more faithful reconstruction at early time steps without the need to lower $\sigma_{\min}$ too much. We further did a grid search with the following hyperparameters, with $s_0 = 2$ and $s_1 = 256$:

- $\sigma_{\min} = [0.01, 0.05, 0.1, 0.2]$;

- $\sigma_{\max} = [1, 1.5, 2, 3, 4, 5]$;

- $\rho = [3, 5, 7]$.

These experiments were run with dropout probability 20%. Afterwards, we experimented with dropout rates $[0\%, 10\%, 20\%, 30\%]$ for the best model with $\sigma_{\min} = 0.05$, $\sigma_{\max} = 3$ and $\rho = 7$, and found 20% dropout rate to work best. We use these hyperparameters also on MNIST and CelebA 64 without further tuning. For the $\beta$-VAE baseline we tuned $\beta$ with the same settings, and searched with the values $\beta = [0.05, 0.1, 0.5, 1.5, 2]$, and found $\beta = 0.1$ to work best (while $\beta = 0.5$ worked best for MNIST).

## A.5 Multistep sampling

To find the optimal time step for multi-step sampling, we first try all the available steps after training, and select the one that gives the best 2-steps FID. We then repeat the procedure for 3 and 4 steps, keeping fixed the time steps found at the previous iteration. While this might not be optimal for more than 2 sampling iterations, we believe it can already provide a good enough heuristic for finding good multi-step sampling times. For MNIST, we use $t = 0.8538$ (idx=162) for 2-steps. For CIFAR-10 and CelebA 64 we test 2, 3 and 4 steps, corresponding to 3, 5 and 7 NFEs, and report the FID results and corresponding time steps in

table 4. While increasing the sampling steps results in lower FID, the improvements decrease as we use more sampling iterations. Perhaps counterintuitively, in some cases adding an extra sampling iteration achieves improved results when re-adding noise at a bigger time step than the iteration before.

| Model, Data | Time steps | Indexes ($\in [1, 257]$) | FID |
|---|---|---|---|
| CoVAE, CIFAR-10 | [1.412, 0.6745, 0.7266] | [198, 146, 151] | [14.06, 13.35, 13.01] |
| CoVAE w/ $\mathcal{L}_{adv}$, CIFAR-10 | [2.4343, 2.3447, 2.0397] | [240, 237, 226] | [9.82 , 9.19, 8.83] |
| CoVAE w/ $\mathcal{L}_{adv}$, CelebA 64 | [1.9376, 2.1193, 2.0659] | [222, 229, 227] | [7.15, 6.98, 6.82] |

Table 4: Multistep FID results and corresponding time steps and time indexes. The results are reported as [2-steps, 3-steps, 4-steps] in the corresponding lists.

## A.6 Data processing

For all the datasets we rescale the values to be in the range [-1,1]. For CIFAR-10 and CelebA 64 we also apply random horizontal flip with 50% probability. For CelebA 64, we first take the center crop of size $148 \times 148$ and then resize to $64 \times 64$ as done in Xiao et al. (2021).

## A.7 Ablation on CIFAR-10

As a base model, we select an architecture with 32.8M parameters, latent size 1024, and batch size 128. By varying these factors in isolation, we can see the effect of each on the generative performance, measured in FID. The remaining hyperparameters are the same as outlined in Appendix A. We report the results of the ablation in figure 6, where we note how CoVAE greatly benefits from bigger latent size, while also improving as batch size and model parameters increase. In the bottom-right plot, we compare the effect of different losses. The first two bars labeled "L2" and "ph" correspond to training the model without boundary condition from equation 24, and with L2 and pseudo-huber loss respectively. The others are a combination of the two, where for example "L2+ph" means that the average decoder network $\hat{x}_{\theta}$ is trained with L2 loss and the ovreall decoder $\mathcal{D}_{\theta}$ is trained with pseudo-huber loss. Note that using only the pseudo-huber loss like in the second column usually leads to instabilities, and the model diverges during training.

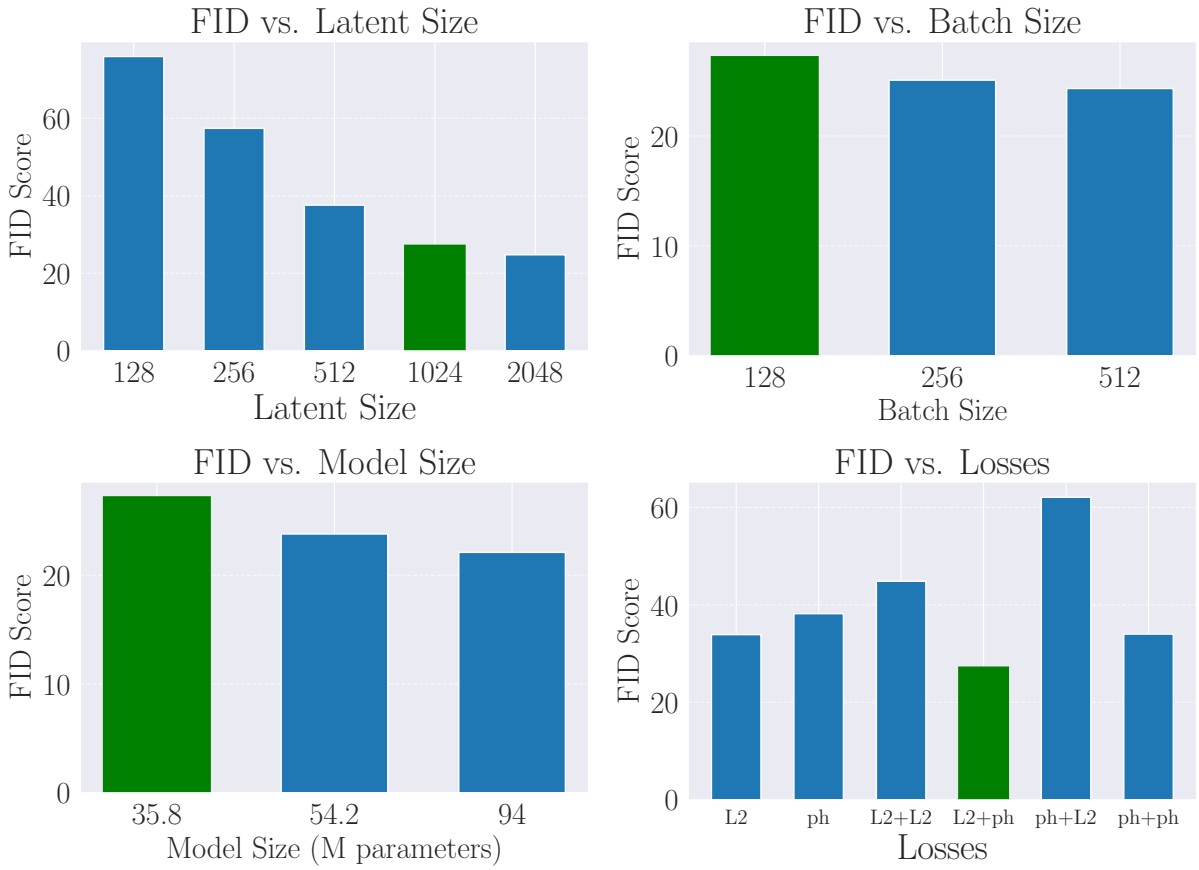

Figure 6: Visualization of the 1-step FID performance (lower is better) for CoVAE with varying hyperparameters. The green bar corresponds to the same run.

# B Qualitative Results

## B.1 Latent space visualization for MNIST

The CoVAE model trained on MNIST has a latent size spatially organized as a $7 \times 7$ grid. This allows us to visualize the latents as grayscale images and get a visual understanding of the learned latent dynamics. In figure 7 we show the learned mean and standard deviation for some of the training images, while varying the time step of the embedding. We further pair each image with a latent noise mask, and show the corresponding sample from the encoded distribution. At small time steps, the encoded means resembles a downscaled version of the input images, and the standard deviations are generally small, resulting in samples indistinguishable from the means. As the time step increases, the mean values get closer to zero and the standard deviations closer to one, resulting in posterior samples which are almost identical to the noise mask. Note that it is not necessary for each encoded distribution at time $t = \sigma_{\max}$ to perfectly match the prior, i.e. isotropic Gaussian, but as in VAEs, we need the aggregate posterior to recover the prior.

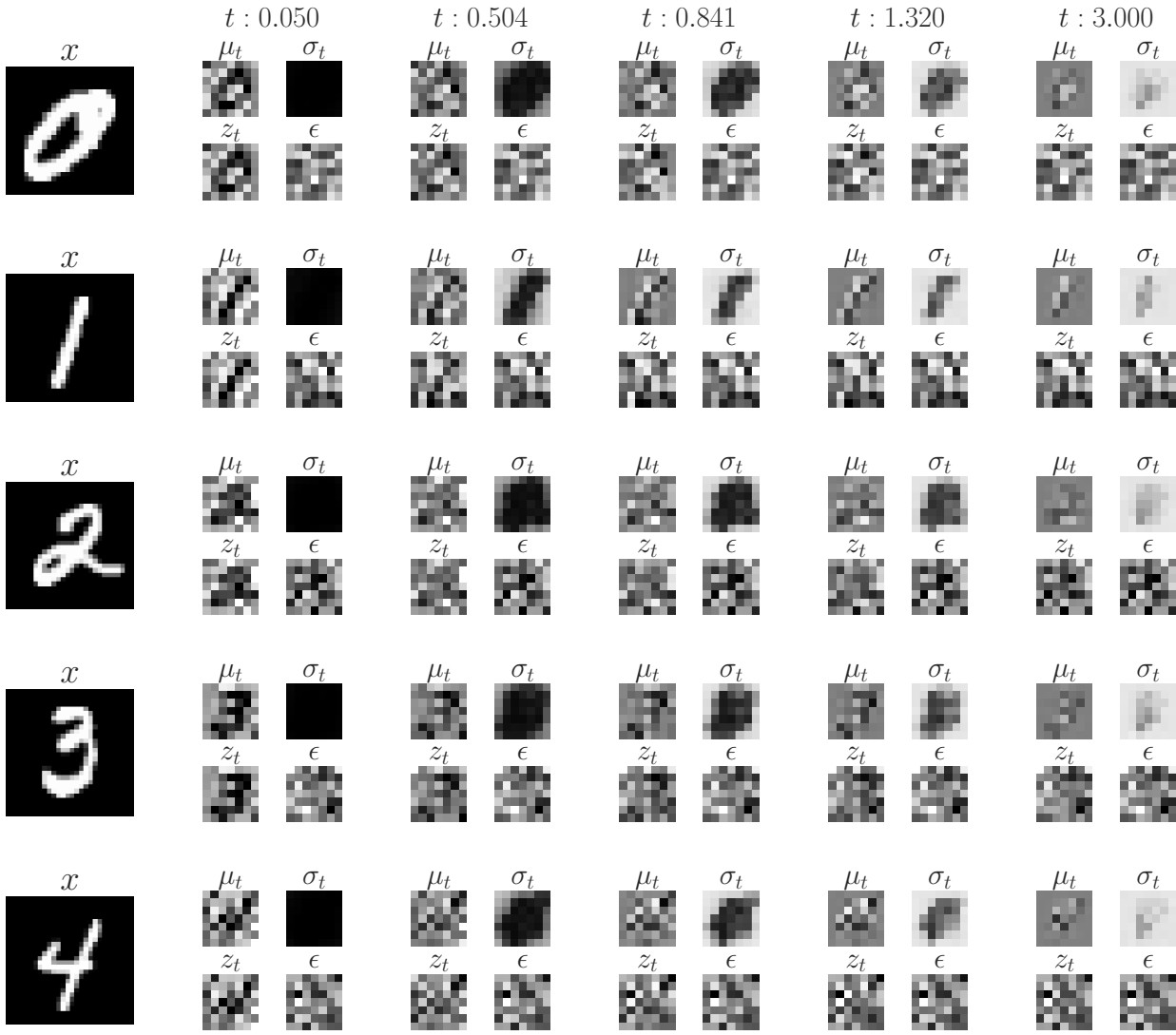

Figure 7: Visualization of latent space learned by CoVAE for different time steps.

## B.2 Generated samples

We report here additional samples from our models, on MNIST in figure 8, CIFAR-10 in figures 9 and 10, and on CelebA 64 in figures 11. Zoom in for best results.

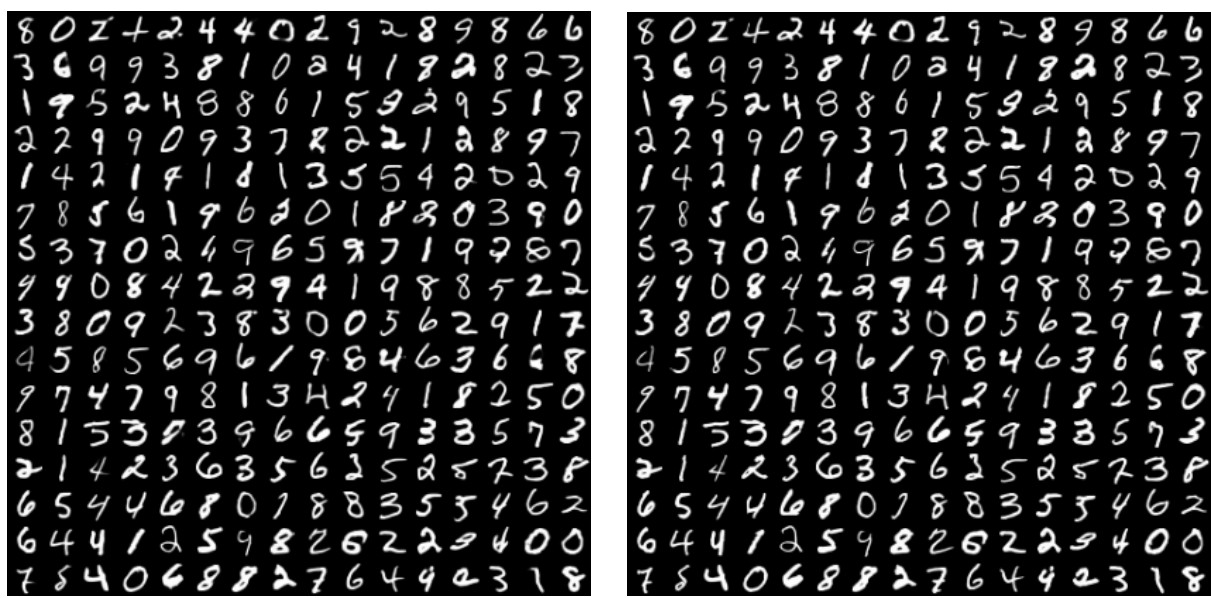

Figure 8: 1-step (FID=5.62, left) and 2-step (FID=3.83, right) generation from CoVAE on MNIST.

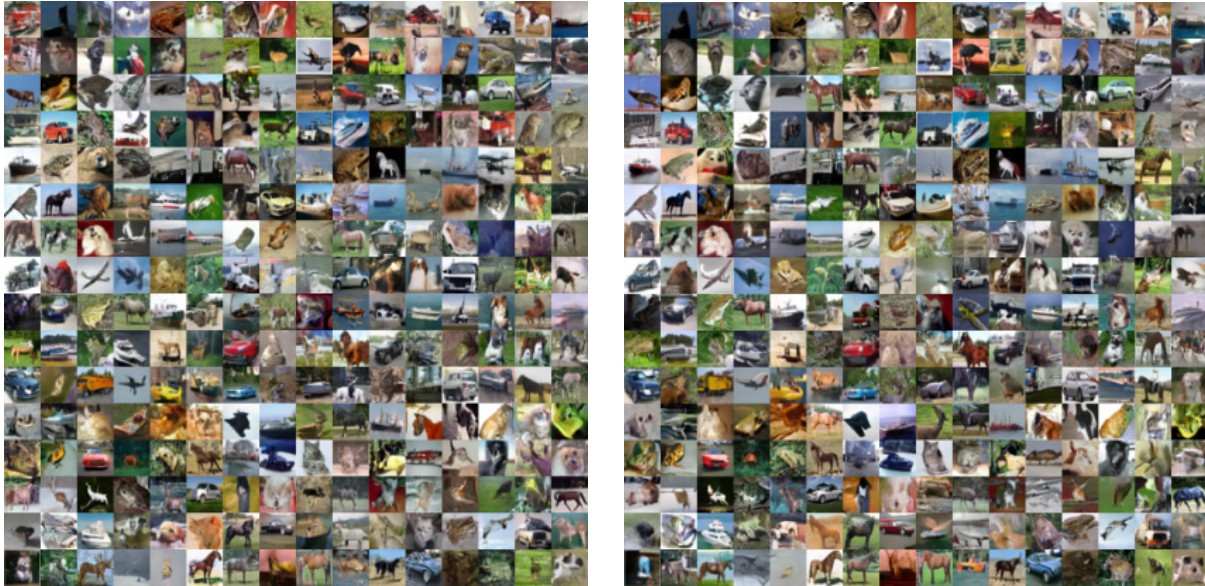

Figure 9: 1-step (FID=17.21, left) and 2-step (FID=14.06, right) generation from CoVAE on CIFAR-10.

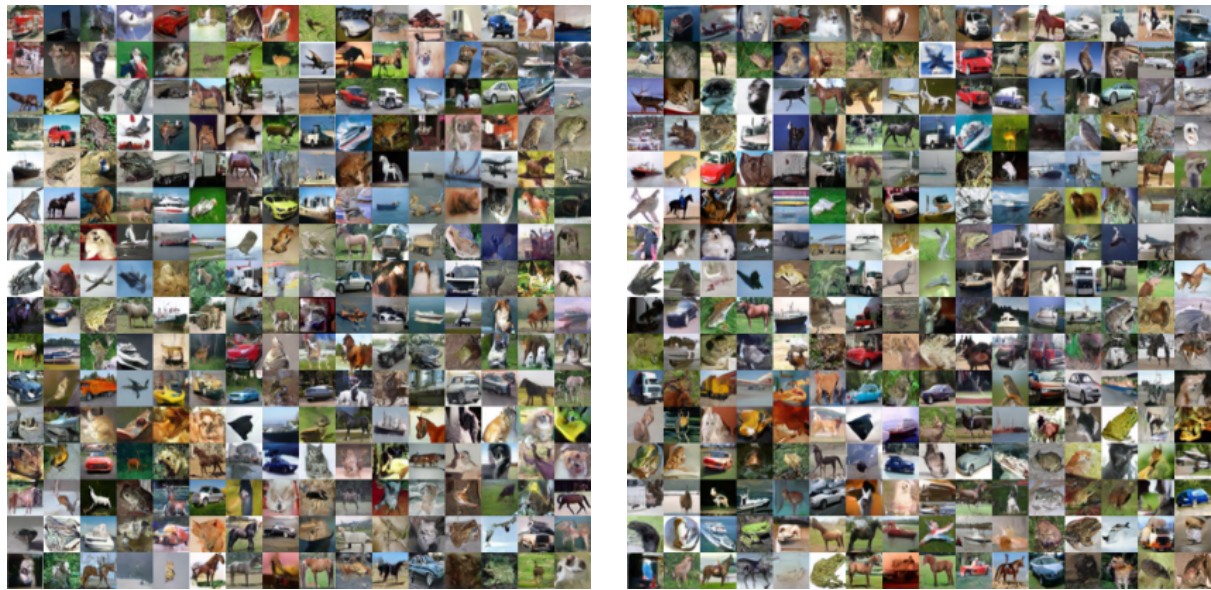

Figure 10: 1-step (FID=11.69, left) and 2-step (FID=9.82, right) generation from CoVAE w/ $\mathcal{L}_{\text{adv}}$ on CIFAR-10.

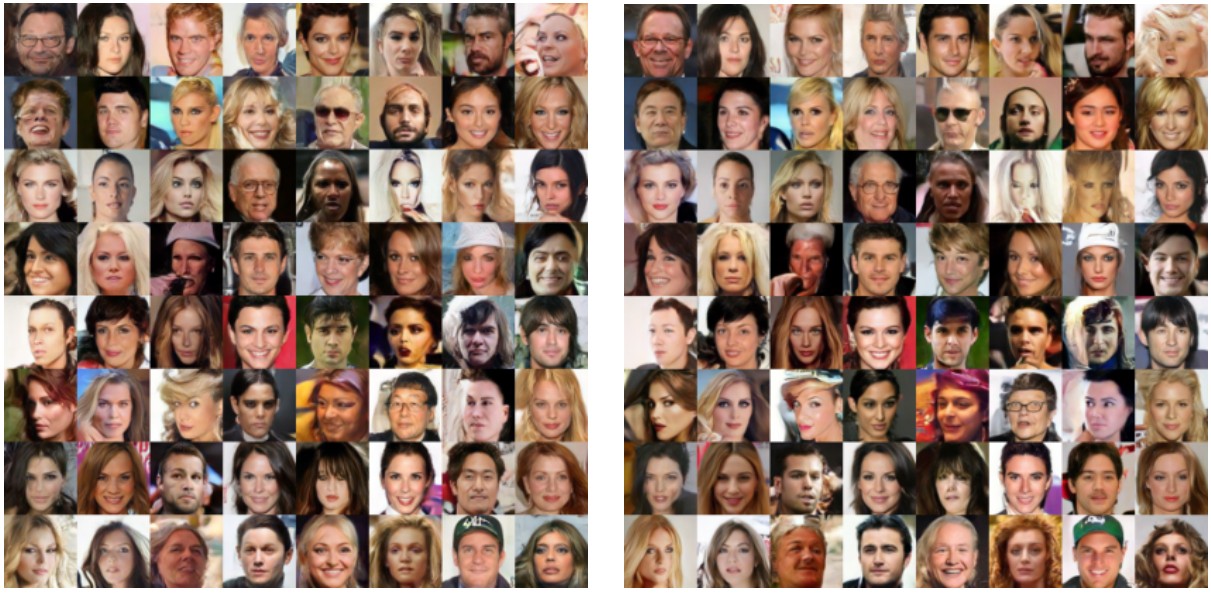

Figure 11: 1-step (FID=8.27, left) and 2-step (FID=7.15, right) generation from CoVAE w/ $\mathcal{L}_{\text{adv}}$ on CelebA 64.

### B.3 Latent interpolation on CelebA 64

In this section, we analyze the effect of image latent space interpolation at different time steps, where a scalar value $\alpha$ is used to interpolate between latent vectors $\boldsymbol{z}_0$ and $\boldsymbol{z}_1$ sampled from the embeddings of two different images $\boldsymbol{x}_0$ and $\boldsymbol{x}_1$ with the same random direction $\boldsymbol{\epsilon}$. We show the reconstructions from the interpolations for different $\alpha$ and different time steps in figures 12 and 13. For small time steps, while the reconstructions without interpolation are almost perfect, we can notice overlapping of the two original images for intermediate values of $\alpha$, especially for images with very distinct features. As $t$ increases, the reconstructions get further away from the original input, but the interpolations transition smoothly between the two images, indicating better latent space disentanglement.

### B.4 Latent manipulation on CelebA 64

Similarly to what done in other VAE works such as Parmar et al. (2021); Pandey et al. (2022), we show some results from latent space manipulation using CelebA 64, as it has 40 annotated binary attributes per image. To add or remove one of such attributes, we first compute an estimate of the latent direction $\boldsymbol{z}_a$ of that attribute by encoding $N$ images with the attribute and $N$ without, sampling from the respective latent spaces, obtaining $\boldsymbol{z}_p$ for positive latent vectors and $\boldsymbol{z}_n$ negatives, and then subtracting the respective means as:

$$\boldsymbol{z}_a = \frac{1}{N} \sum (\boldsymbol{z}_p) - \frac{1}{N} \sum (\boldsymbol{z}_n), \tag{30}$$

where we set $N = 100$. The modified latent of an encoded image that does not have the selected attribute is computed with the following:

$$\boldsymbol{z}' = \boldsymbol{z} + \psi \boldsymbol{z}_a, \tag{31}$$

where $\psi$ is a scalar that regulates the strength of the update. Similarly, to remove an attribute one can simply subtract $\boldsymbol{z}_a$ instead. As CoVAE can obtain latent representations at different time steps, we show the effect of latent manipulation at different time steps and for different manipuation strength in figures 14 and 15, obtained with CoVAE /w $\mathcal{L}_{\mathrm{adv}}$. For the modifications at small time steps to become visible, a bigger $\psi$ is needed, which also seems to introduce some artifacts, but can obtain a faithful reconstruction to the original image. For bigger time steps, the modifications tend to be more visible already with small $\psi$, and increasing $\psi$ has less visible artifacts, but the reconstructed image is further away from the original input.

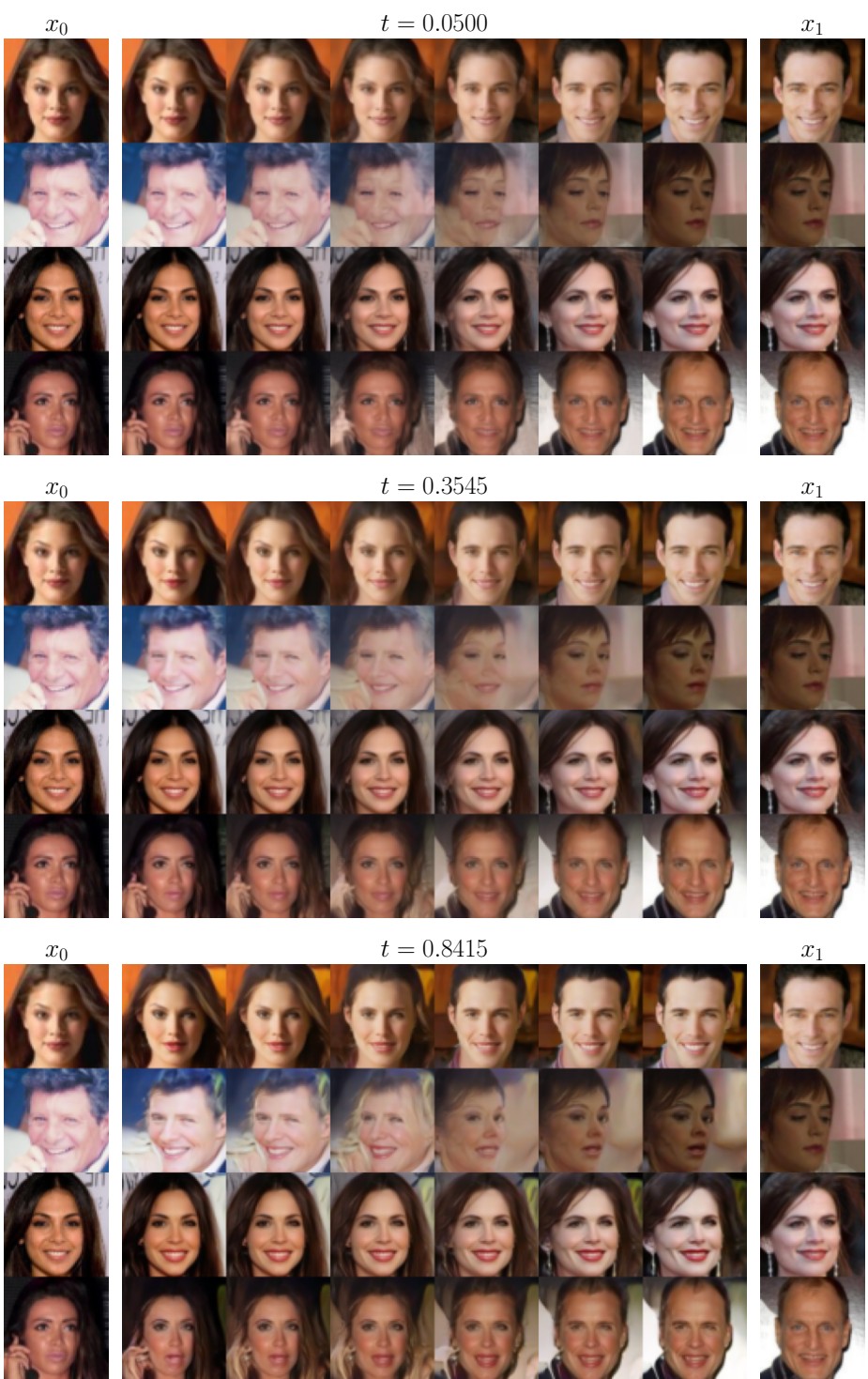

Figure 12: The figure shows the reconstruction from latent space interpolation between two data points displayed on the right and left hand side columns. The interpolations are obtained with mixing factor $\alpha \in [0, 0.2, 0.4, 0.6, 0.8, 1]$ from left to right in the central grid. The embeddings are obtined with time step $t$ displayed on top.

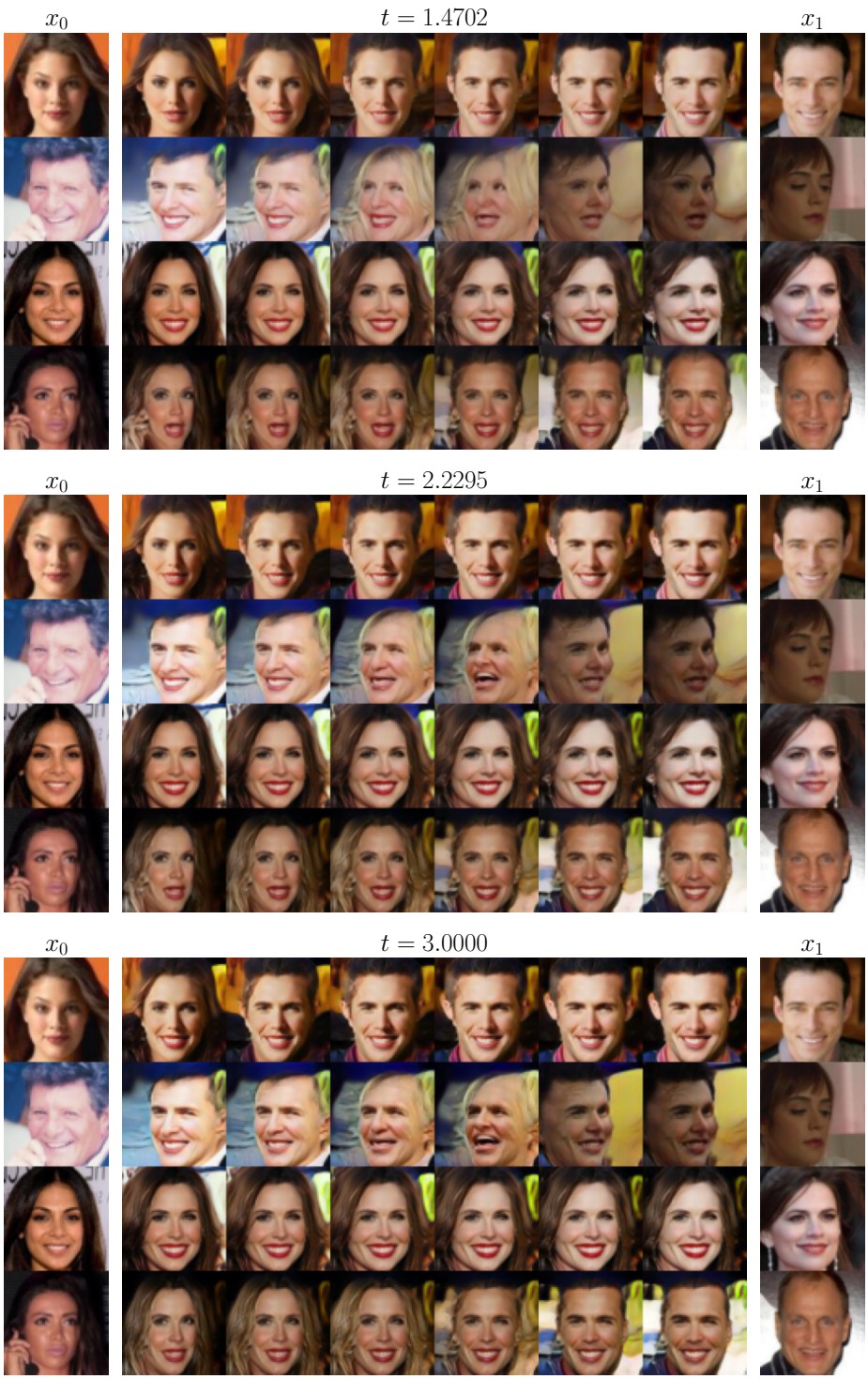

Figure 13: Continuation of figure 12 for higher values of t.

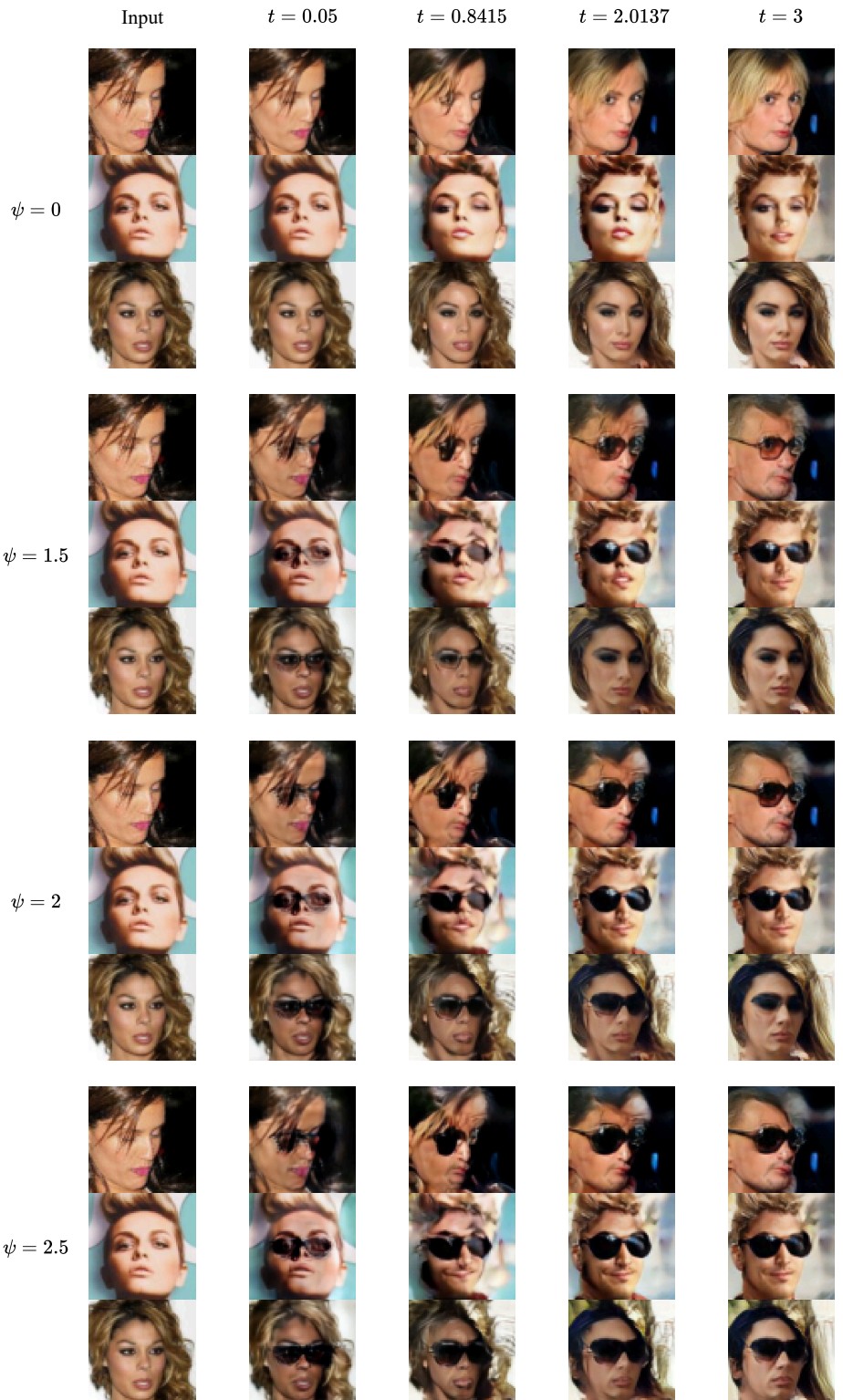

Figure 14: Latent space manipulation experiments adding the latent direction for the attribute "Eyeglasses". The first row with $\psi = 0$ corresponds to no manipulation, and is used to show the reference reconstructed embedding.

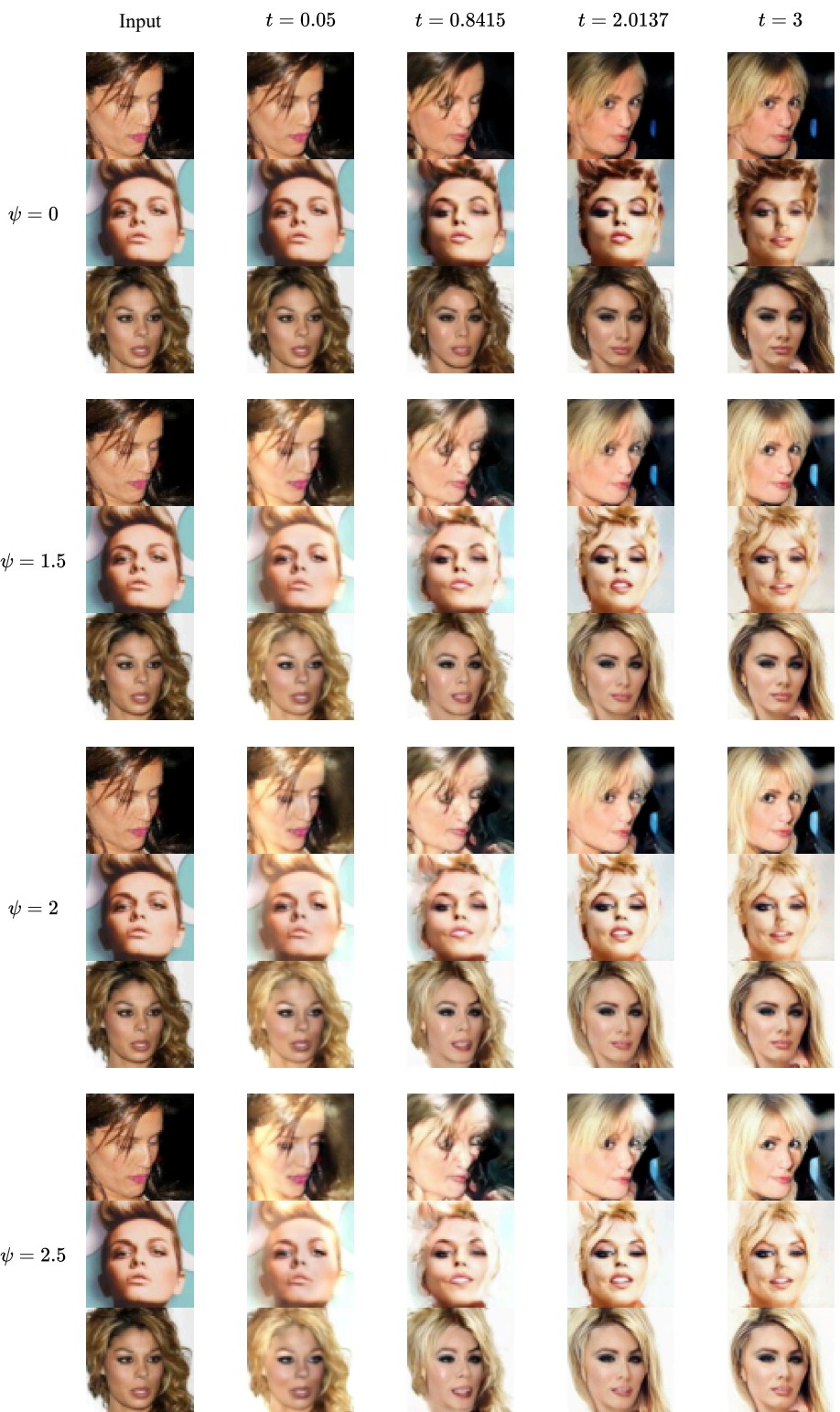

Figure 15: Latent space manipulation experiments adding the latent direction for the attribute "Blonde". The first row with $\psi = 0$ corresponds to no manipulation, and is used to show the reference reconstructed embedding.

# C  Alternative formulations for CoVAE

In this section, we discuss alternative formulations of CoVAE. We first present a simplified variant that reduces the reliance on time-dependent regularization while preserving the core autoencoding structure. We then describe an extension of CoVAE to discrete data. While these variants do not match the performance of the full CoVAE model on simple benchmarks, we believe they represent interesting directions within the proposed framework, and we leave their scaling and further refinement to future work.

## C.1  Simplified CoVAE

In its original formulation, CoVAE learns a time-dependent latent embedding whose noise level is implicitly controlled by regularization schedules. While effective, this design relies on the choice of weighting functions $\lambda(t)$ and $\beta(t)$, as well as careful tuning of $\sigma_{\min}$ and $\sigma_{\max}$. Moreover, the resulting signal-to-noise ratio (SNR) across time does not admit a closed-form expression, which complicates principled design choices.

Here, we show that the CoVAE formulation can be simplified by decoupling the encoder from explicit time dependence and introducing a fixed latent-space corruption process. In particular, we define a time-dependent latent variable by combining a time-independent encoder with an explicit noise injection in latent space:

$$z_t = a_t \mathcal{E}_\phi(x) + b_t \epsilon, \quad \epsilon \sim \mathcal{N}(0, I), \tag{32}$$

where $a_t$ and $b_t$ are scalar functions controlling the noise level. A simple choice is $a_t = 1$ and $b_t = t$, which progressively increases the noise variance in latent space.

With this formulation, the latent distribution transitions toward a Gaussian without requiring explicit $\lambda(t)$ and $\beta(t)$ schedules. However, since the encoder is no longer regularized through a variational objective, an additional constraint is required to prevent unbounded latent embeddings. To this end, we introduce a latent regularization term of the form $\gamma \|\mathcal{E}_\phi(x)\|^2$ added to the reconstruction loss in Equation 25, where $\gamma$ is a scalar hyperparameter ($\gamma = 0.001$ in our experiments). This regularization is equivalent to imposing a fixed-variance Gaussian prior on the latent representation.

Alternatively, to further simplify the model and remove the need for additional hyperparameters, we normalize the encoder output using LayerNorm (Ba et al., 2016) followed by a tanh activation. The resulting simplified CoVAE (s-CoVAE) is trained using the same consistency-based objective as CoVAE, but is computationally more efficient, as the encoder is evaluated only once per training example and the latent corruption is applied directly to the encoded representation at different noise levels.

Since the latent corruption process is explicitly defined, we can reuse standard preconditioning functions for the decoder parametrization in Equation 24, with a minor modification to account for the different definition of the average decoder. In practice, we set $c_{\text{skip}}(t) = 1$ and retain the remaining scaling functions unchanged. Similarly to CoVAE, we include an average-decoder loss scaling factor $\lambda_d(t) = \sigma_{\text{data}}^2 / (t^2 + \sigma_{\text{data}}^2)$.

We evaluate s-CoVAE on CIFAR-10 using the same experimental setup as in Appendix A.7 (32.8M parameters, latent size 1024, batch size 128). The variant using $\gamma$ regularization achieves a one-step FID of 40.42, while the normalization-based variant performs slightly better with a one-step FID of 38.18. In comparison, the full CoVAE model achieves a one-step FID of 27.21, likely due to the additional flexibility provided by the learned time-dependent latent regularization. Nevertheless, s-CoVAE represents a viable alternative when training efficiency and simplicity are prioritized.

### C.1.1  Derivation of the Average Denoiser in VE Diffusion

We consider the Variance Exploding (VE) forward process:

$$\mathcal{F}(x, t) = x + t\epsilon, \quad \epsilon \sim \mathcal{N}(0, I) \tag{33}$$

Assume the data distribution is $x \sim \mathcal{N}(0, \sigma_{\text{data}}^2 I)$. We want to compute the posterior mean:

$$\hat{x}_t = \mathbb{E}[x \mid x_t] \tag{34}$$

This is a Gaussian denoising problem: we observe $\boldsymbol{x}_t$ which is a noisy version of $\boldsymbol{x}$. Since both $\boldsymbol{x}$ and the noise are Gaussian, the posterior $p(\boldsymbol{x} \mid \boldsymbol{x}_t)$ is also Gaussian. Let:

$$p(\boldsymbol{x}) = \mathcal{N}(\boldsymbol{x}; 0, \sigma_{\mathrm{data}}^2 \boldsymbol{I})$$
$$p(\boldsymbol{x}_) \mid \boldsymbol{x}) = \mathcal{N}(\boldsymbol{x}_t; \boldsymbol{x}, t^2 \boldsymbol{I})$$

Then by Bayes' rule:

$$p(\boldsymbol{x} \mid \boldsymbol{x}_t) \propto p(\boldsymbol{x}_t \mid \boldsymbol{x}) p(\boldsymbol{x}) \tag{35}$$

This is a standard case of Gaussian conjugate priors. The posterior mean is given by:

$$\hat{\boldsymbol{x}}_t = \left( \frac{1}{\sigma_{\mathrm{data}}^2} + \frac{1}{t^2} \right)^{-1} \cdot \left( \frac{\boldsymbol{x}_t}{t^2} \right) \tag{36}$$

Simplifying:

$$\hat{\boldsymbol{x}}_t = \frac{\sigma_{\mathrm{data}}^2}{\sigma_{\mathrm{data}}^2 + t^2} \boldsymbol{x}_t \tag{37}$$

### C.1.2 Variance and Standard Deviation

The variance of the posterior mean across samples $\boldsymbol{x}_t$ is:

$$\mathrm{Var}(\hat{\boldsymbol{x}}_t) = \left( \frac{\sigma_{\mathrm{data}}^2}{\sigma_{\mathrm{data}}^2 + t^2} \right)^2 \cdot \mathrm{Var}(\boldsymbol{x}_t) \tag{38}$$

Since:

$$\mathrm{Var}(\boldsymbol{x}_t) = \mathrm{Var}(\boldsymbol{x} + t\boldsymbol{\epsilon}) = \sigma_{\mathrm{data}}^2 + t^2 \tag{39}$$

We have:

$$\mathrm{Var}(\hat{\boldsymbol{x}}_t) = \frac{\sigma_{\mathrm{data}}^4}{\sigma_{\mathrm{data}}^2 + t^2} \tag{40}$$

And therefore, the standard deviation is:

$$\mathrm{Std}(\hat{\boldsymbol{x}}_t) = \frac{\sigma_{\mathrm{data}}^2}{\sqrt{\sigma_{\mathrm{data}}^2 + t^2}} \tag{41}$$

### C.1.3 Boundary conditions with average denoiser

The boundary conditions commonly used in CMs are of the form:

$$f_\theta(\boldsymbol{x}_t, t) = c_{\mathrm{skip}}(t)\boldsymbol{x}_t + c_{\mathrm{out}}(t)\boldsymbol{F_\theta}(\boldsymbol{x}_t, t) \tag{42}$$

with $c_{\mathrm{skip}}(\sigma_{\min}) = 1$ and $c_{\mathrm{out}}(\sigma_{\min}) = 0$. For the VE case, $c_{\mathrm{skip}}(t)$ is defined as:

$$c_{\mathrm{skip}}(t) = \frac{\sigma_{\mathrm{data}}^2}{t^2 + \sigma_{\mathrm{data}}^2} \tag{43}$$

The variable $c_{\mathrm{skip}}(t)$ is used to multiply $\boldsymbol{x}_t$ which has a standard deviation of $\sqrt{t^2 + \sigma_{\mathrm{data}}^2}$, so that $\mathrm{STD}(c_{\mathrm{skip}}(t)\boldsymbol{x}_t) = \frac{\sigma_{\mathrm{data}}^2}{\sqrt{\sigma_{\mathrm{data}}^2 + \sigma^2(t)}}$. As we can see, this is already equivalent to the standard deviation of the average denoiser $\hat{\boldsymbol{x}}_t$, so the new boundary condition is simply:

$$f_\theta(\boldsymbol{x}_t, t) = \hat{\boldsymbol{x}}_t + c_{\mathrm{out}}(t)\boldsymbol{F_\theta}(\boldsymbol{x}_t, t), \tag{44}$$

where in practice we approximate the average decoder/denoiser with a neural network.

## C.2 CoVAE with discrete data

Unlike diffusion and consistency based models that operate directly in ambient space, CoVAE learns a data-dependent mapping to a latent representation, which naturally supports both continuous and discrete data. While the use of CoVAE with discrete data is not the focus of this work, we report a proof-of-concept experiment on binary MNIST. In this setting, we replace the $\ell_2$ and pseudo-Huber reconstruction losses with a binary cross-entropy loss.

Figure 16 shows one-step samples from CoVAE trained using the same hyperparameters as in Section 4, except for $\sigma_{\min}$, which is increased to 0.5. This configuration achieves a one-step sampling FID of 0.58 and a reconstruction FID of 0.17. While this preliminary experiment does not allow us to assess scalability to more complex discrete domains such as text or biological data, it suggests that extending CoVAE to discrete generative settings is a promising direction for future work.

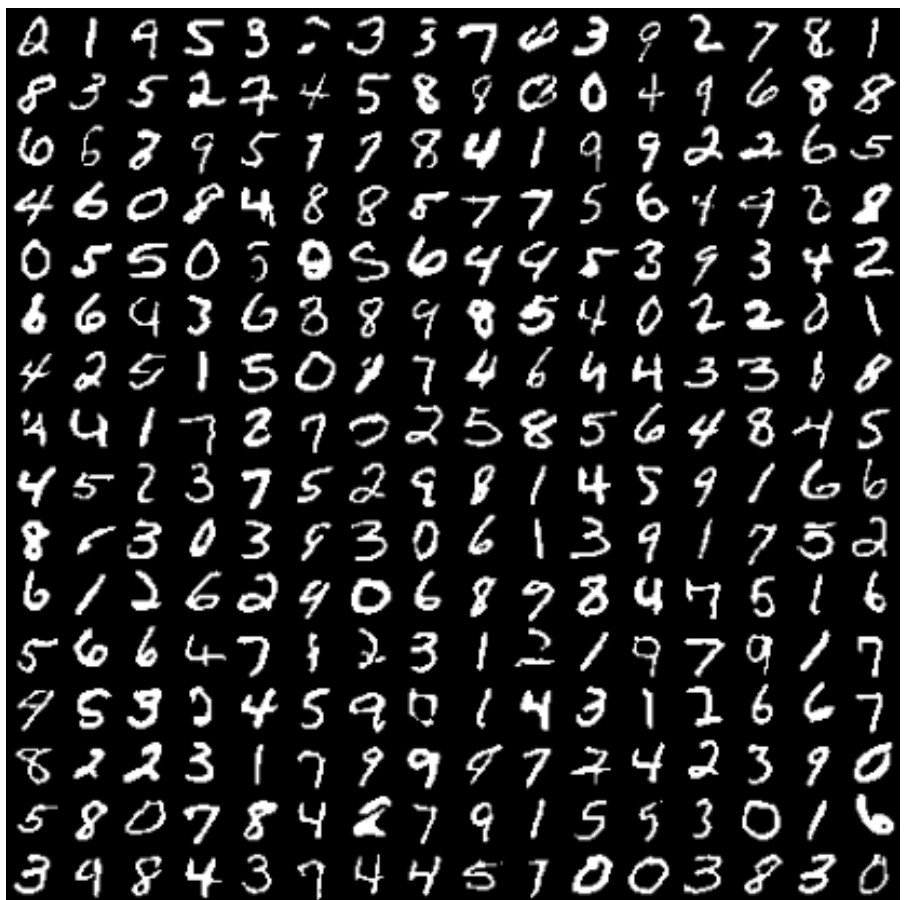

Figure 16: 1-step samples from binary MNIST, FID=0.58.

