# OpenReview forum: "CoVAE: Consistency Training of Variational Autoencoders"
_TMLR — Under review for TMLR_

### Review · Reviewer_PMEz · 2026-07-12

**Summary Of Contributions:**

The paper proposes CoVAE, a novel generative model that introduces a continuous, time-indexed latent distribution, diverging from conventional VAEs that rely on a single prior. To optimize CoVAE, the authors replace the standard pointwise reconstruction loss with a consistency loss between decoder outputs at different timesteps. The framework also incorporates an empirical method to better satisfy boundary conditions in the latent space. Evaluations on MNIST, CIFAR-10, and CelebA 64 demonstrate improved generative performance over standard single-stage VAE baselines.

Strengths

- Proposes a novel integration of time-dependent VAE regularization and consistency training to enable single-stage, one-step or few-step generation.
- Effectively mitigates the prior hole problem common in conventional VAEs, improving generation without relying on a cumbersome two-stage training pipeline or auxiliary priors.

Weaknessess

- Crucial experiments evaluating the reconstruction-generation tradeoff and the overall quality of the learned representations are missing.
- Discussions regarding computational cost and scalability are insufficient, leaving the practical limitations of the single-stage architecture unclear.
- The theoretical motivation, particularly the interpretation for the consistency objective in Section 3.2, lacks depth and clarity.

**Audience:**

Yes

**Audience Explanation:**

The paper extends the standard VAE framework by proposing a novel training objective that integrates consistency models. Researchers working on deep generative models will likely be interested in the empirical findings demonstrating that this formulation improves standard VAE generative performance.

**Claims And Evidence:**

No

**Claims Explanation:**

The central claim is that CoVAE supports one-step or few-step generation directly from the latent prior while maintaining strong reconstruction performance and a structured latent space. However, this claim is not fully supported due to some missing experiments and ablations. Please see the requested changes for details.

**Requested Changes:**

Experiments

- Include a comparison against beta-VAEs with varying regularization strengths, specifically providing a reconstruction-generation tradeoff plot to empirically support the claim that CoVAE mitigates this inherent tradeoff.
- Provide a more comprehensive latent analysis beyond the provided qualitative t-SNE visualizations to prove the latent space is well-structured. For example, a linear probing experiment as done in DC-VAE could be included.

Motivations and Interpretations

- Expand the motivation behind employing the consistency objective. Justify why this specific objective is used and how it mechanically enables single-stage training. A discussion regarding latent collapse (as CoVAE is single-stage) would also be helpful.
- Explicitly state the standard boundary and smoothness assumptions mentioned.
- Section 3.2 requires refinement.
    - The current interpretation should more clearly explain the theoretical underpinnings of the framework, moving beyond stating immediate relationships.
    - Address the theoretical limitation regarding the reconstruction loss upper bound at t=0, as the current bounding formulation from Equation 18 appears very loose.
    - The paragraph on "Time-dependent rate constraints" would benefit from clarification. Rather than primarily detailing equation manipulations, providing a substantive discussion on the resulting conclusions would add significant value.

Computational Cost

- Provide a comparison and in-depth discussion regarding the computational cost. Single-stage approaches typically induce high VRAM usage when scaled to higher resolutions compared to decoupled methods like latent diffusion models.
- Given the use of H100 GPUs in the experiments, either scaling to high-resolution datasets or discussing the real-world applicability and efficiency limitations of CoVAE would be beneficial for readers to properly understand the framework's tradeoffs.

---

> ### Author Response · Authors · 2026-07-15
>
> We thank the reviewer for the positive assessment of the novelty of CoVAE and for the constructive suggestions. We agree that several aspects of the presentation can be improved and will revise the manuscript accordingly.
>
> ### Reconstruction-generation trade-off
>
> We agree that illustrating this trade-off is valuable. In the submitted paper, we report the best-performing β-VAE baseline obtained after tuning β. During our experiments, decreasing β consistently improved reconstruction quality while substantially degrading generative performance, whereas increasing β beyond the reported value degraded both reconstruction and generation. We therefore selected the strongest β-VAE baseline rather than reporting a full sweep. We will clarify this experimental procedure in the revised manuscript.
>
> ### Latent representation analysis
>
> Beyond the t-SNE visualizations, the paper already includes latent interpolation experiments illustrating the structure of the learned latent space. We agree that additional analyses, such as linear probing, could provide further characterization. However, we believe such experiments would primarily evaluate representation quality rather than the central contribution of the paper, namely the proposed consistency-based training objective. We also note that CoVAE already substantially outperforms DC-VAE on both generation and reconstruction metrics, suggesting that the proposed training mechanism yields a stronger generative autoencoder.
>
> ### Motivation for the consistency objective and latent collapse
>
> The consistency objective is the core contribution of CoVAE and plays a role analogous to standard consistency models, with the key difference that the latent dynamics are induced by the learned encoder rather than by an analytically defined forward diffusion process in ambient space. Rather than reconstructing the input independently at every regularization level, the decoder progressively bootstraps decoding information from informative latent states toward increasingly prior-aligned ones.
>
> Furthermore, latent collapse is mitigated through the time-dependent regularization schedule, which preserves informative latent representations at small timesteps while gradually enforcing prior alignment as t increases. To further support this interpretation, we provide KL (rate) curves averaged over 10,000 training images for our best models on each dataset:
>
> https://anonymous.4open.science/r/img-D287/combined_kl_rate.png
>
> These curves show that the expected KL decreases smoothly and approaches zero as t increases, validating the intended progression from informative to prior-aligned latent representations.
>
> ### Theoretical presentation
>
> We appreciate the suggestions regarding Section 3.2. Our goal was not to derive a new ELBO, but to provide a pathwise interpretation of the CoVAE objective. We agree that this section can be made clearer by explicitly stating the smoothness assumptions, discussing the interpretation of the rate constraints in greater depth, and clarifying that the reconstruction bound is intended as an interpretive upper bound rather than a tight optimization guarantee. We will revise this section accordingly.
>
> ### Computational cost and scalability
>
> We agree that scalability deserves additional discussion. Since memory usage depends strongly on the hardware, software stack, and implementation details, we chose to report GFLOPs as a hardware-independent measure of computational complexity. Nevertheless, we will expand the discussion of computational trade-offs and explicitly acknowledge that scaling CoVAE to high-resolution datasets remains an important direction for future work. We will also discuss the practical trade-offs between single-stage generative autoencoders and decoupled latent-generation pipelines in the revised manuscript.

---

### Review · Reviewer_3TZc · 2026-07-13

**Summary Of Contributions:**

CoVAE is a single-stage generative autoencoder that learns a time-dependent continuum of latent distributions, ranging from near-deterministic encodings to Gaussian-aligned latents, and enforces decoder consistency across neighboring regularization levels. This enables one- or few-step generation without a separate latent prior while improving generation and reconstruction FID over VAE baselines on MNIST, CIFAR-10, and CelebA-64. The paper also provides a pathwise interpretation of the objective rather than presenting it as a new ELBO.

**Audience:**

Yes

**Audience Explanation:**

The paper is valuable to general audience working in generative models and specifically generation-driven VAEs.

**Claims And Evidence:**

Yes

**Claims Explanation:**

All claims are backed by sufficient experiments and are convincing

**Requested Changes:**

My main concern with this paper is that the proposed method is only validated on small-scale or low-resolution datasets, such as CIFAR-10 and CelebA-64. Nowadays, it is increasingly important to report results on moderately sized datasets such as ImageNet-256, especially for a new generative architecture like this one. Many GAN/VAE-based methods become very difficult to train or converge on ImageNet-1K, even though they work well on CIFAR-10 or CelebA. I understand that training on ImageNet-1K can be costly, but I would expect the authors to at least report preliminary results and discuss any convergence difficulties. This could help further distiguish this work with previous VAE efforts. Methodology-wise, I'm quite convinced and think it's a very interesting attempt to revisit VAE-base generative arch with CM objective.

---

> ### Author Response · Authors · 2026-07-15
>
> We thank the reviewer for the positive assessment and for highlighting the potential value of evaluating CoVAE on larger-scale datasets. We agree that demonstrating scalability to more challenging settings is an important direction for future work.
>
> At present, our experiments were constrained by a limited computational budget, and we were therefore only able to perform the experiments reported in the submission. As a result, we could not conduct additional ImageNet-scale training within the review period.
>
> We would also like to point out that many recent works reporting ImageNet-256 results (e.g., latent diffusion and latent consistency models) do not train an autoencoder directly on ImageNet-256 pixels. Instead, they operate in the latent space of a separately pretrained high-capacity autoencoder, effectively evaluating the second-stage generative model rather than the autoencoder itself. In contrast, CoVAE is a single-stage generative autoencoder that jointly learns both the latent representation and the generative mapping from scratch, making this setting substantially different from the standard latent ImageNet benchmarks.
>
> We nevertheless agree that evaluating CoVAE on larger-scale datasets is an important next step. We will explicitly discuss this limitation in the revised manuscript and identify large-scale experiments as a key direction for future work.

---

### Review · Reviewer_y7xo · 2026-07-13

**Summary Of Contributions:**

This paper studies consistency training for VAEs. Its main goal is to approximate, within a single training stage, the functionality that is typically divided between an autoencoder and a separately trained latent generative model in two-stage latent-generation pipelines. CoVAE learns a time-conditioned family of VAE posteriors, where larger timesteps correspond to stronger KL regularization and better alignment with the Gaussian prior. Neighboring timesteps are coupled by enforcing consistency between their decoded outputs. The latent samples are obtained from the same input image and the same reparameterization noise, which defines a coupled stochastic path across timesteps. At inference, the model can directly decode a prior sample at a large timestep, or refine the generated image by re-encoding and decoding it at additional timesteps. Quantitatively, CoVAE improves over the reported VAE-family baselines.

## Strengths

- The central idea is technically interesting and appears distinct from standard VAE training. Defining multiple regularization levels within a shared latent space and connecting them through iterative generation provides an intuitive link between VAEs and recent diffusion or consistency-based methods. The proposed latent transition may offer a useful way to combine prior sampling and reconstruction within a single model.


- The reported results improve over the evaluated VAE-family methods. CoVAE consistently achieves better generation metrics than the standard VAE and $\beta$-VAE baselines, and also compares favorably with several existing single-stage VAE models in the reported settings.

## Weaknesses

- The significance of the method as an image-generation framework remains unclear. Although the paper motivates CoVAE through comparisons with diffusion and consistency models, its generation quality remains substantially below that of modern GAN, diffusion, flow, and consistency-model families. The decoder is still trained primarily through reconstruction objectives with a compact latent space, which may fundamentally limit sample fidelity. The large improvement obtained by adding an adversarial loss further suggests that the proposed VAE objective alone may be insufficient for high-quality synthesis. It is therefore unclear whether replacing an expressive latent prior with a single-stage generative autoencoder offers a practical advantage over existing two-stage pipelines.

- The training objectives impose a strong and insufficiently analyzed tension. For the same input image, the encoder produces timestep-dependent posterior distributions, and latent samples generated using the same noise are encouraged to decode to consistent outputs. At the same time, each posterior is regularized toward the Gaussian prior, and the auxiliary decoder is directly trained to reconstruct the input. Strong prior alignment at large timesteps reduces the information available about the input, whereas reconstruction and cross-timestep consistency require preserving that information. These objectives are not strictly incompatible at finite regularization, but the paper does not explain or measure how the model balances them. Moreover, the average decoder and residual output share most of the decoder network rather than forming fully independent decoders, so it is unclear whether the residual branch learns a meaningful consistency correction or whether the model mainly behaves as a collection of timestep-conditioned VAEs.

- The multi-step inference procedure is not validated at the latent-distribution level. The intended interpretation appears to be that an initial sample generated from a strongly regularized latent can be re-encoded at a less regularized timestep and moved toward a more realistic reconstruction regime. However, the paper does not measure whether the posterior obtained by re-encoding generated images overlaps with the posterior obtained from real images at the same timestep. Rather, visualized examples shows that there are significant gap between images from different timesteps.
This, distances between the posterior distributions or images from different timesteps would be needed to verify so that re-encoding actually moves generated samples toward latent regions occupied by real data. Without such evidence, the multi-step improvement may simply result from generic encode–decode projection rather than the proposed consistency mechanism.

- The ablation and latent-path analyses are insufficient to isolate the contribution of the proposed components. A direct comparison with a time-dependent $\beta$-VAE using ordinary reconstruction is particularly important. The paper should also evaluate the model without decoded-output consistency, without the residual branch, and with independent rather than shared reparameterization noise. In addition, the divergence or overlap between neighboring timestep posteriors should be measured directly. The current experiments about image-space consistency results lacks to confirm the effectiveness of the proposed method.

**Audience:**

Yes

**Audience Explanation:**

Yes. The proposed connection between time-dependent VAE regularization and consistency training is technically interesting and is likely to be relevant to at least part of the TMLR audience working on generative modeling, variational autoencoders, and one- or few-step generation. The reported improvements over VAE-family baselines further suggest that the approach is worth investigating.

However, the empirical analysis is currently insufficient to fully understand the proposed mechanism. In particular, the paper does not directly verify whether the consistency objective produces a coherent transition across timestep-dependent latent representations or preserves consistent decoded outputs along the corresponding path. More detailed analyses of cross-timestep latent overlap, decoded-output consistency, and ablations against a time-dependent VAE without consistency training would substantially strengthen the findings. Thus, the idea itself is of potential interest to the TMLR community, even though the current experiments do not yet provide a complete validation of its central interpretation.

**Broader Impact Concerns:**

Although its about the generative modeling, current model does not seem to have concerns about ethical implications.

**Claims And Evidence:**

No

**Claims Explanation:**

No. The quantitative results provide convincing evidence that CoVAE outperforms the evaluated VAE and $\beta$-VAE baselines. However, the experiments do not sufficiently support the paper’s central mechanistic claims. In particular, the paper does not compare CoVAE with the time-dependent $\beta$-VAE introduced in Equation 8, nor does it isolate the effects of consistency loss. Therefore, while the empirical improvement over VAE-family baselines is clear, it remains uncertain whether the gains specifically arise from the proposed consistency-training mechanism.

The latent-path and multi-step sampling interpretations are also not directly validated. The paper does not measure cross-timestep posterior overlap, decoded-output consistency along a fixed shared-noise trajectory, or whether re-encoded generated samples occupy latent regions similar to those of real images. Consequently, the reported multi-step improvement may partly reflect generic encode–decode projection rather than the proposed latent consistency path. Overall, the performance claims relative to the evaluated VAE baselines are supported, but the broader claims about the learned consistency mechanism and its role in multi-step generation are not yet supported by sufficiently convincing evidence.

**Requested Changes:**

- Isolate the effect of consistency training with a matched time-dependent $\beta$-VAE baseline.
The authors should train the model defined in Equation 8 using the same architecture, timestep-dependent regularization, latent dimensionality, and training budget as CoVAE, but with an ordinary reconstruction loss at each timestep. Comparing its one-step generation and reconstruction performance with CoVAE is necessary to determine whether the reported gains arise from cross-timestep consistency, rather than from time conditioning and jointly learning multiple VAE regularization regimes.

- Directly demonstrate that CoVAE learns a coherent cross-timestep path.
For fixed input images and shared reparameterization noise, the paper should visualize decoded outputs across the complete timestep trajectory and quantitatively measure their consistency, for example using pixel- or perceptual-space distances between neighboring and distant timesteps. The authors should also characterize the divergence, overlap, or sample-wise displacement of the corresponding latent posteriors. The current timestep-specific latent visualizations show that the representations change with t, but do not establish that the consistency objective produces a smoothly coupled latent path whose decoded content remains stable.

- Verify that multi-step improvement is specific to the proposed consistency mechanism.
The identical re-encode–decode sampling procedure should be applied to the matched time-dependent $\beta$-VAE baseline, with generation metrics reported after each step. The authors should additionally compare the latent distributions obtained by encoding generated and real images at the same inference timestep. These experiments would clarify whether re-encoding moves generated samples toward real-data latent regions, or whether the observed improvement is primarily a generic autoencoder-projection effect that does not depend on consistency training.

---

> ### Author Response · Authors · 2026-07-15
>
> We thank the reviewer for the detailed evaluation and for recognizing both the novelty of the proposed formulation and the empirical improvements over VAE-family methods. We appreciate the constructive suggestions. Below we address the main concerns.
>
> ### 1. Objective of CoVAE and the role of the consistency loss
>
> The reviewer raises several concerns regarding the interaction between KL regularization, reconstruction, and consistency. We believe this stems from a misunderstanding of the training objective.
>
> Importantly, the CoVAE decoder is not directly trained to reconstruct the input image. Rather, its primary objective is to match the prediction produced at a neighboring, less regularized latent state. Only the auxiliary average decoder is trained with a reconstruction objective, and it is used solely to construct the boundary parametrization inspired by consistency models. The consistency decoder itself never minimizes a reconstruction loss against the data.
>
> Similarly, the apparent tension between stronger KL regularization and reconstruction is precisely the motivation behind CoVAE. The proposed consistency objective is designed to progressively transfer decoding information from informative, weakly regularized latent states toward increasingly prior-aligned latent states. This bootstrapping mechanism is the central idea of the paper and is discussed both intuitively and through the pathwise interpretation in Section 3.2.
>
> Regarding the shared average decoder and residual branch, if the residual branch simply collapsed into an ordinary timestep-conditioned VAE decoder, one would expect predictions at highly regularized timesteps to converge toward the conditional expectation, producing the characteristic blurry reconstructions of standard VAEs. Empirically, this is not what we observe, suggesting that the residual branch indeed learns nontrivial consistency corrections.
>
> ### 2. Requests for a time-dependent β-VAE baseline and removing consistency
>
> The reviewer requests comparisons against the model defined in Eq. (8), as well as variants without decoded-output consistency.
>
> Equation (8) is intentionally introduced only as a conceptual intermediate step to motivate CoVAE. It simply amortizes a collection of independent β-VAEs sharing parameters. Without coupling neighboring timesteps through the consistency objective, each timestep still faces the original reconstruction-versus-prior trade-off independently, and there is no mechanism to transfer reconstruction information toward prior-aligned latent states. Consequently, Eq. (8) is not expected to solve the problem that motivates the paper.
>
> For the same reason, removing the decoded-output consistency loss does not constitute an ablation of CoVAE, but simply recovers the time-dependent β-VAE introduced in Eq. (8). The consistency objective is the defining component of CoVAE, and removing it fundamentally changes the learning objective rather than isolating one component of the proposed method.
>
> ### 3. Shared-noise trajectories and latent-path interpretation
>
> The reviewer requests analyses based on coherent cross-timestep trajectories, posterior overlap, and measuring distances along a latent path.
>
> Here we would like to clarify an important point. Sharing the same reparameterization noise does not imply that samples lie along a true latent ODE trajectory. The shared noise is introduced for the same reason as in discrete consistency models: it couples neighboring stochastic latent samples so that the consistency loss compares corresponding perturbations rather than unrelated latent draws. This coupling is sufficient for defining the discrete consistency objective, but it should not be interpreted as recovering an exact latent trajectory.
>
> Consequently, visualizing reconstructions obtained by simply fixing the noise and varying the timestep would not recover the latent path optimized by CoVAE. In fact, as the posterior changes with increasing regularization, such a procedure is expected by construction to move between different latent representations and therefore produce different decoded images.
>
> Accordingly, our theoretical analysis is based on a stochastic pathwise interpretation induced by shared-noise coupling rather than on explicit latent ODE trajectories. We will revise the manuscript to make this distinction clearer, as we agree that this interpretation could be stated more explicitly.
>
> Similarly, improvements obtained through multi-step sampling should not be interpreted as projecting samples back onto a recovered latent trajectory, but rather as repeatedly applying the learned encoder-decoder consistency mapping at progressively less regularized latent states.

---

> ### Author Response · Authors · 2026-07-15
>
> ### 4. Comparison with modern diffusion models
>
> Finally, the reviewer notes that CoVAE does not match the image quality of state-of-the-art diffusion or consistency models.
>
> We fully agree and do not intend to make such a claim. The goal of CoVAE is different: to investigate whether consistency training can be integrated directly into a single-stage generative autoencoder, avoiding the separate latent prior used by modern two-stage pipelines while substantially improving over conventional VAE-family methods. Throughout the experimental section, comparisons are therefore restricted to single-stage VAE-based baselines, which we believe constitute the appropriate point of comparison.